# The divisome but not the elongasome organizes capsule synthesis in *Streptococcus pneumoniae*

Rei Nakamoto [1], Sarp Bamyaci [2], Karin Blomqvist[2,3], Staffan Normark[2], Birgitta Henriques-Normark [2,3] & Lok-To Sham [1] ✉

The bacterial cell envelope consists of multiple layers, including the peptidoglycan cell wall, one or two membranes, and often an external layer composed of capsular polysaccharides (CPS) or other components. How the synthesis of all these layers is precisely coordinated remains unclear. Here, we identify a mechanism that coordinates the synthesis of CPS and peptidoglycan in *Streptococcus pneumoniae*. We show that CPS synthesis initiates from the division septum and propagates along the long axis of the cell, organized by the tyrosine kinase system CpsCD. CpsC and the rest of the CPS synthesis complex are recruited to the septum by proteins associated with the divisome (a complex involved in septal peptidoglycan synthesis) but not the elongasome (involved in peripheral peptidoglycan synthesis). Assembly of the CPS complex starts with CpsCD, then CpsA and CpsH, the glycosyltransferases, and finally CpsJ. Remarkably, targeting CpsC to the cell pole is sufficient to reposition CPS synthesis, leading to diplococci that lack CPS at the septum. We propose that septal CPS synthesis is important for chain formation and complement evasion, thereby promoting bacterial survival inside the host.

To produce two identical daughter cells, various cell envelope layers must be synthesized and divided in a highly coordinated manner. This will ensure both siblings have a fully fortified wall to cope with challenges from the ever-changing environment[1–3]. Coordination between peptidoglycan (PG) and outer membrane synthesis has been recently reported in *Escherichia coli*[4], where outer membrane proteins are preferentially inserted at sites where new PG is synthesized. By comparison, it remains unclear how the synthesis of secondary wall polymers like the capsular polysaccharide (CPS) synchronizes with the rest of the cell envelope biogenesis. A link between CPS and PG synthesis was suggested because the serine-threonine kinase PknB (or StkP in *Streptococcus pneumoniae*) negatively regulates CPS synthesis in *Staphylococcus aureus*[5]. Controlling CPS synthesis is especially important during cytokinesis, where all cell envelope layers are remodeled at the same rate[1–3]. Otherwise, lesions on the cell

surface may result in cells that are susceptible to antibody and complement attacks[6–8].

In *Streptococcus pneumoniae* (the pneumococcus), CPS synthesis is spatiotemporally orchestrated by the bacterial tyrosine kinase (BY-kinase) system CpsCD[9–11]. CpsC is a membrane protein that belongs to the polymerase co-polymerase type 2b (PCP2b) family. It interacts with and presumably stimulates the activity of the polymerase CpsH, thereby regulating the length of capsule polymers[9]. Autophosphorylation of the cognate cytoplasmic kinase CpsD reduces the polymer length likely by modulating the activity of CpsC[9]. The regulatory system is then reset by CpsD dephosphorylation by the cognate phosphatase CpsB[12–14]. CpsC may also integrate CPS synthesis with chromosome segregation, as illustrated by the nucleoid defects associated with the CpsD^{Y→F} mutation[10,15]. However, this link is likely dependent on the growth medium or the genetic background because

[1]Infectious Diseases Translational Research Programme and Department of Microbiology and Immunology, Yong Loo Lin School of Medicine, National University of Singapore, Singapore 117545, Singapore. [2]Department of Microbiology, Tumor and Cell Biology (MTC), Karolinska Institutet, Stockholm SE-17177, Sweden. [3]Clinical Microbiology, Karolinska University Hospital Solna, SE-17176 Stockholm, Sweden. ✉e-mail: lsham@nus.edu.sg

we could not detect it in the serotype 2 strain D39W[9]. Several reports also indicated that CpsC and CpsD are found at the septum[10,11,16]. Yet, how they arrive there remains unclear. It probably involves PG synthases[9], chromosome segregation factors[10], and/or the serine-threonine kinase StkP[5], but this has not been tested experimentally.

In most bacteria, PG is synthesized by two distinct complexes called the elongasome (or the Rod complex) and the divisome[17–22]. Central to the divisome is the cytoskeletal protein FtsZ, a structural homolog of tubulin. FtsZ recruits cell division proteins such as FtsA and FtsQLB[23–26]. It also positions the SEDS (shape, elongation, division, and sporulation) family protein FtsW and an associated class B penicillin-binding protein (bPBP) at the septum[27]. Together, FtsW and the partner bPBP produce the glycan strands and peptide crosslinks that form the septal PG. The elongasome mirrors the divisome as it also contains the SEDS family protein RodA and another cognate bPBP[28]. Synthesis of septal and peripheral PG is regulated by FtsQLB[29,30] and MreCD[31], respectively. Losing these regulators results in aberrant PG formation that changes the cell shape, often leading to non-viable cells. In *S. pneumoniae* and related ovococci that lack MreB, both the elongasome and the divisome are located at the division septum. They are close to each other and therefore cannot be easily resolved, especially at the early phase of the cell cycle when the two complexes colocalize[27,28,32,33]. Thus, it remains enigmatic whether CPS synthesis tracks with the synthesis of septal PG, peripheral PG, or both.

Here, we demonstrated that disruption of the divisome but not the elongasome assembly abolishes septal CpsC localization. Inactivation of *cpsC* is sufficient to disperse other CPS proteins, whereas deletions of the rest of the *cps* genes had little effect on the positioning of CpsC. This result suggests that CpsC is upstream in the recruitment pathway. Remarkably, targeting CpsC to the cell pole is sufficient to re-route CPS synthesis. The inability to synthesize CPS at the septum sensitizes the cell towards complement deposition, indicating that the location of the CPS complex is important for virulence. Together, we propose a mechanism that could explain how the pneumococcal cell is covered by the protective polysaccharide capsule.

## Results

### CpsC is at the site of PG and CPS synthesis

To study the coordination of CpsC with PG and CPS synthesis in pneumococci, we constructed a C-terminal superfolder green fluorescence protein (sfGFP) fusion of CpsC at the native locus (*cpsC-L-sfgfp*). As disruption of CpsC is lethal in the wild-type serotype 2 strain D39W[9], the strain was made in a genetic background where *cpsE* is under the control of an inducible $P_{Zn}$ promoter (Δ*cpsE* // $P_{Zn}$-*cpsE*). CpsE is the initiating phosphoglycosyltransferase[34] and upon its depletion, capsule production is reduced, rendering CpsC dispensable[9]. This strain construction strategy avoids potential suppressor mutations from accumulating in the transformants. *cpsC-L-sfgfp* remained functional because CPS synthesis was restored to the wild-type level when $Zn^{2+}$ was added to the medium (Supplementary Fig. 1A, B). Also, we did not detect significant CpsC-sfGFP degradation by immunoblotting (Supplementary Fig. 1C). Expectedly, CpsC-sfGFP was detected at midcell. Pulse-chase experiments using fluorescent D-amino acids (FDAA) and capsule immunostaining confirmed that CPS synthesis, PG synthesis, and CpsC are co-localized (Fig. 1). As the length of the cell correlates with its age, PG incorporation apparently precedes CpsC recruitment (Fig. 1). However, the CPS and PG complexes cannot be well resolved under standard light microscopy or 3D structural illumination microscopy (3D-SIM) described in our previous study[9]. Thus, we conclude that the enzymes required for CPS and PG biogenesis are recruited to the septum, including the CpsCD complex. Yet, we were unsure whether CPS synthesis tracks with septal PG synthesis, peripheral PG synthesis, or both.

### Recruitment of CpsC to the septum requires the divisome but not the elongasome

Next, we investigated how CpsC is recruited to midcell. In *S. pneumoniae*, the two PG synthesis complexes are organized by FtsZ[17]. Regulated by guanosine triphosphate (GTP) binding and hydrolysis, FtsZ polymerizes and tethers to the cell membrane by FtsA to form a ring-like structure, in which FtsZ manifests as numerous short filaments that rotate around the division annulus[35]. Other cell division proteins are then recruited to the ring-like structure. These proteins perform several essential functions for cytokinesis such as DNA translocation[36], PG synthesis[29,37], and cell wall remodeling[23,38–41]. We hypothesize that if CPS synthesis is recruited to the septum by any of these factors, depletion of FtsZ will disperse the CPS complex and delocalize CpsC. To deplete FtsZ, we inserted a $P_{Zn}$-*ftsZ* cassette into an ectopic locus and deleted the native copy of *ftsZ*. If $Zn^{2+}$ was not supplemented to the culture, cells became rounded, enlarged, and ultimately lysed (Fig. 2A and supplementary Fig. 2A). After FtsZ was deprived, the CpsC-sfGFP signal dispersed around the cell. This result indicates that the FtsZ-ring is required to localize CpsC. In *S. pneumoniae*, placement of the FtsZ-ring is thought to be guided by MapZ (or LocZ)[42,43]. However, Δ*mapZ* did not delocalize CpsC (Fig. 2C), perhaps because other fail-safe mechanisms such as RocS[15], CcrZ[44], and EzrA[45] can substitute for its function. Indeed, while Δ*rocS* and Δ*ccrZ* did not affect CpsC positioning (Fig. 2C), EzrA depletion dispersed CpsC-sfGFP (Fig. 2B). EzrA is a positive regulator of FtsZ-ring formation in pneumococci[45]. It also serves as an adaptor protein by interacting with several cell division proteins like FtsA, StkP, Pbp1a, and ZapJ, thereby bridging cell division and PG synthesis[45]. Together, our results suggest that early FtsZ-ring assembly is required to localize CpsC to the septum.

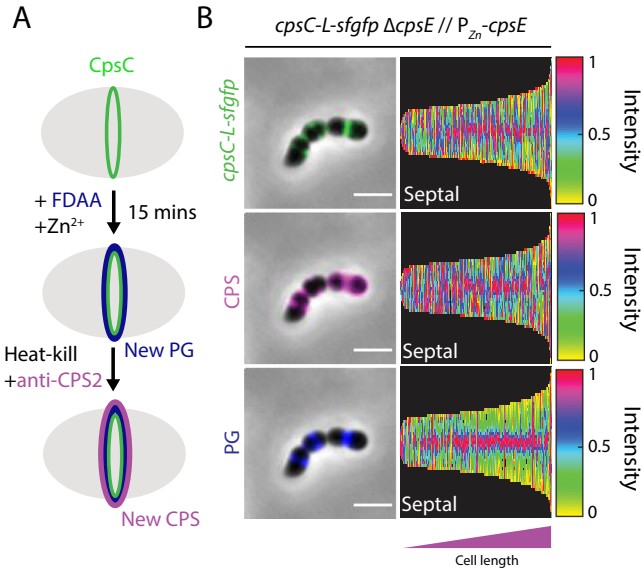

**Fig. 1 | CpsC-L-sfGFP, peptidoglycan (PG) synthesis, and capsular polysaccharide (CPS) synthesis colocalize at the septum. A** Strain NUS0882 [*cpsC-L-sfgfp* Δ*cpsE* // $P_{Zn}$-*cpsE*] was grown until OD$_{600}$ reached 0.1. Capsule production was induced by adding $Zn^{2+}$ and the nascent PG synthesis was tracked by adding FDAA. After 15 min, cells were heat-killed, immunolabeled with anti-serotype 2 capsule antibodies (anti-CPS2), and detected using fluorescent secondary antibodies. The cells were then visualized by fluorescence microscopy. **B** Shown are the representative micrographs (left) and the corresponding demographs (right) of the fluorescent channels. CpsC-L-sfGFP, anti-CPS2, and FDAA signals in the micrographs were pseudo-colored green, pink, and blue, respectively. For demographs, newly divided, shorter cells are on the left and the septated, longer cells are on the right. Bar, 2 μm. Demograph analysis includes data from three biologically independent experiments (*n* = 500).

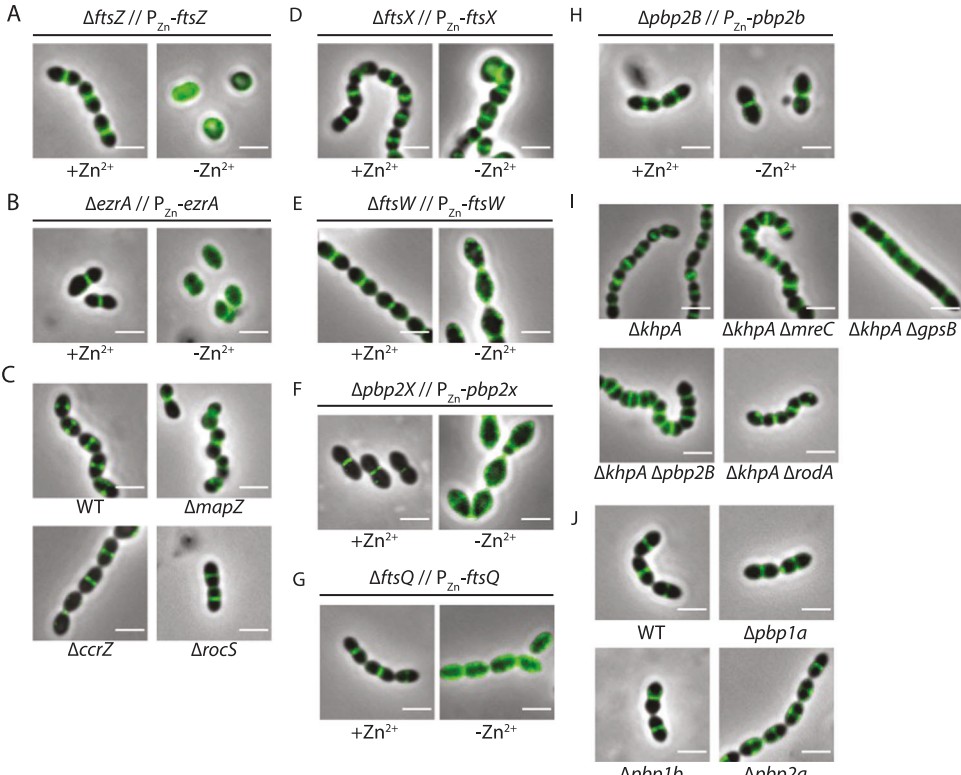

**Fig. 2 | The CPS complex is recruited by the divisome but not the elongasome.** Strains harboring CpsC-L-sfGFP with the indicated genetic modifications (Supplementary Table S1) were grown until $OD_{600}$ reached 0.2. To deplete the indicated proteins (**A**, **B**, **D–H**), cells were collected by centrifugation, washed once with an equal volume of BHI broth without $Zn^{2+}$ supplement, and diluted to an $OD_{600}$ of 0.1 in the same medium. Growth was resumed by incubation at 37 °C in 5% $CO_2$. Cultures were mounted on a glass slide and observed under fluorescence microscopy when the growth rate began to reduce (Supplementary Fig. S2). As a control, a duplicate set of cultures was prepared but with $Zn^{2+}$ supplementation (labeled as +$Zn^{2+}$). For panels **C**, **I**, and **J**, cells were grown directly in the BHI medium and imaged when $OD_{600}$ reached 0.2. Bar, 2 μm. Similar results were obtained from three biologically independent experiments.

After the FtsZ-ring is formed, the PG synthesis complexes begin to segregate. The matured divisome spirals towards to the lumen of the division annulus, thereby closing the septum[27,46]. At the same time, the peripheral PG complex slowly migrates toward the center[28], expanding the sacculi by separating the old glycan strands and the incorporating new glycan strands[46]. The components in each complex are under intense investigation. To test whether they are important for localizing CpsC-sfGFP, representative proteins in each complex were selected, including a regulator of PG hydrolysis (FtsX), factors for septal PG synthesis (Pbp2x, FtsW, FtsQ), factors for peripheral PG synthesis (Pbp2b, RodA, MreC), class A penicillin-binding proteins (Pbp1a, Pbp1b, and Pbp2a), and a cytosolic adaptor protein GpsB[47,48] that shuttles between the two complexes. The essentiality of GpsB, as well as many peripheral PG synthesis proteins, can be suppressed by *khpA* inactivation[49] or an increase in MurA or MurZ activities[50]. Δ*khpA* leads to FtsA overproduction. As a result, the cell reduces in size and possibly tolerates less sidewall synthesis[49]. Similar observations were made in *E. coli* where FtsZ overproduction suppresses lesions in the Rod complex[51]. On the other hand, the reason why Δ*khpA* can suppress Δ*gpsB* is less clear until recently[50] because it is independent of FtsA overexpression[49]. Thus, this genetic relationship allows us to construct null mutants of *mreC*, *pbp2B*, *rodA*, and *gpsB*. As expected, cells lacking *mreC*, *pbp2B*, and *rodA* appeared smaller and more spherical (Fig. 2I), whereas Δ*gpsB* resulted in large, elongated cells (Fig. 2I), likely because Δ*gpsB* prevents closures of the septal ring by the divisome[47]. Despite the severe phenotypes caused by these mutations, none of them changed the localization of CpsC (Fig. 2). Likewise, depletion of FtsX and deletions of class A PBPs did not affect CpsC recruitment (Fig. 2D, J). FtsX is part of the FtsEX complex that regulates PG hydrolysis during cell division or elongation. Unlike in *Escherichia coli*, FtsEX and its cognate PG hydrolase in gram-positive bacteria are thought to be part of the elongasome[52,53]. Therefore, we conclude that elongasome assembly and function are not required for recruiting CpsC to the septum.

Although several division proteins like EzrA and FtsZ seem to arrive at the septum earlier[27], the exact order of their assembly has not been determined in *S. pneumoniae*[17]. As the pneumococcus does not produce FtsN, the last protein being recruited to the divisome is still an open question[17]. To test which divisome protein is required to recruit CpsC, we constructed depletion strains of *ftsQ*, *pbp2X*, and *ftsW*. Since the onset of lysis in these strains is different from the *ftsZ* depletion strain NUS1659 [*cpsC-L-sfgfp* Δ*ftsZ*::P-*kan-rpsL*+ // $P_{Zn}$-*ftsZ*], we empirically determined the timing to collect cells for imaging (Supplementary Fig. S2). To control for the pleiotropic effects that may arise from the cell shape defects, we confirmed that FtsZ-L-sfGFP remained in place after these proteins were depleted (Supplementary Fig. S3). Similar to disruptions in early FtsZ-ring assembly, CpsC-sfGFP was delocalized after *pbp2X*, *ftsW*, and *ftsQ* depletion (Fig. 2E–G). We next tested if the effect is specific to the sfGFP tag by conducting the same experiment with a *cpsC-L-FLAG₃* strain. CpsC-L-$FLAG_3$ was fully functional, and there was no significant $FLAG_3$-tag cleavage detected. In addition, the expression levels of CpsC remains unchanged when *ftsZ*, *pbp2X*, and *ftsW* were depleted (Supplementary Fig. S3). Like the sfGFP-tagged CpsC (Fig. 3, Supplementary Fig. S3) and down expression of *pbp2x*, *ftsW*, and *ftsQ* also delocalized CpsC-L-$FLAG_3$ (Supplementary Fig. S3). Thus, CpsC recruitment requires late divisome proteins in the divisome.

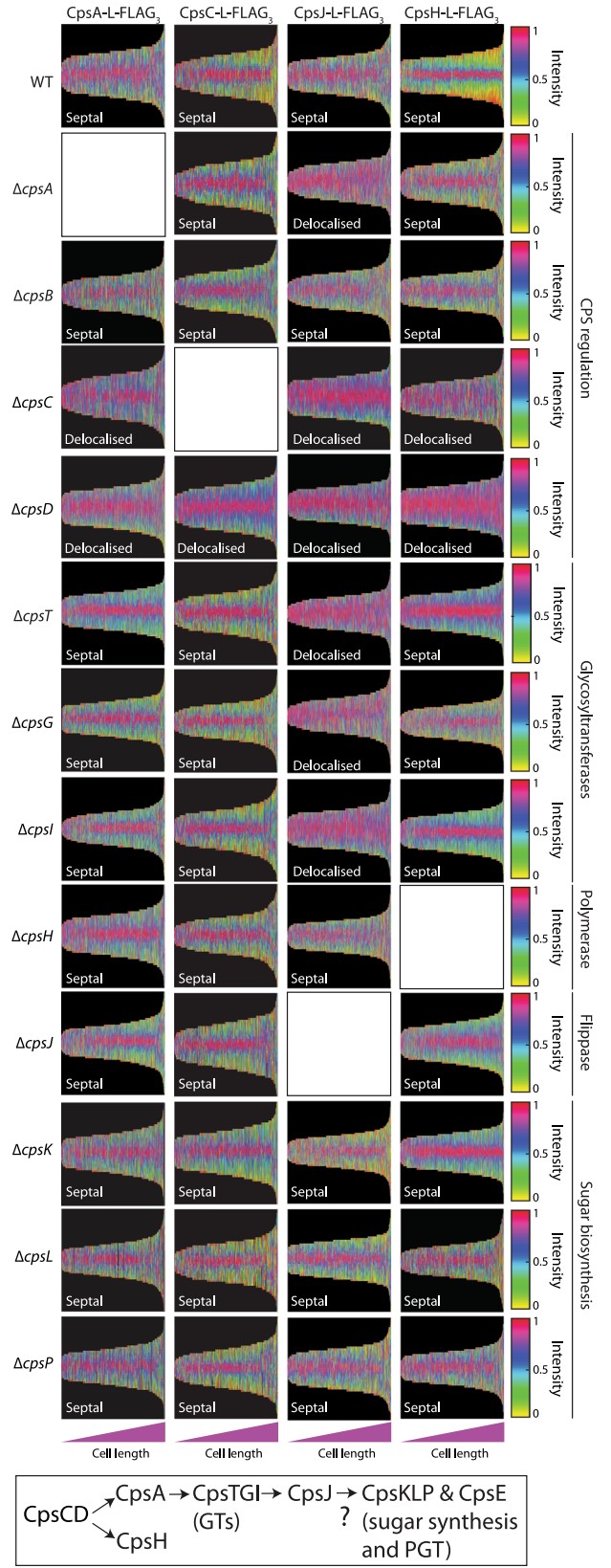

**Fig. 3 | The hierarchy of CPS complex assembly.** Strains harboring the indicated FLAG tag fusions (CpsA-L-FLAG$_3$, CpsC-L-FLAG$_3$, CpsJ-L-FLAG$_3$, and CpsH-L-FLAG$_3$) and the deletion mutants were grown in BHI until the culture reached the mid-logarithmic phase (OD$_{600}$ is between 0.2 and 0.4) (see Supplementary Table 1) in the absence of Zn$^{2+}$. Cells were harvested and fixed on a poly-L-lysine-coated glass slide. They were then immunostained with anti-FLAG as described in "Methods". Shown are the demographs of the fluorescent channels depicting the location of the FLAG tag fusion proteins as well as the tentative recruitment pathway. Representative micrographs are presented in Supplementary Fig. S5. Demograph analysis includes data from three biologically independent experiments ($n = 500$). GT Glycosyltransferase, PGT phosphoglycosyltransferase.

namely CpsT, CpsF, CpsG, and CpsI, sequentially add the sugar residues until the repeating unit is made. The final lipid-linked precursor is transported to the other side of the membrane by the CpsJ(Wzx) flippase, then polymerized by CpsH(Wzy). The CPS polymer is ligated to PG presumably by CpsA[57], although genetic analyses generated mixed results[58] and biochemical reconstitution of this reaction is still pending. In addition, it was shown that CpsC is necessary to recruit CpsA, CpsD, and CpsH to the septum[10,16]. CpsJ is also septal-localized and is likely to be recruited by the CpsCD complex[10], but this has not been tested experimentally.

To determine the localization of CPS enzymes, we constructed C-terminal FLAG-tagged fusions of CpsA, CpsJ, and CpsH. We attempted to fuse other CPS proteins with a FLAG-tag or sfGFP as well. However, these constructs were either non-functional or unstable. CpsA-L-FLAG$_3$, CpsJ-L-FLAG$_3$, and CpsH-L-FLAG$_3$ were functional and localized at the septum (Supplementary Figs. S1A, S1B, S4A and S5C). To determine the recruitment cascade and their interdependency for proper localization, we deleted constituents of the CPS complex and examined if the localization of these proteins would be changed. We propose that CpsC-L-FLAG$_3$ arrives at the divisome first, because Δ*cpsC* delocalized CpsA, CpsJ, and CpsH (Fig. 3, Supplementary Fig. S5C). Furthermore, CpsC localization is independent of other CPS factors, except CpsD, which was shown to stabilize CpsC (Fig. 3; Supplementary Fig. S5A). In addition, induction of *cpsE* did not affect the localization of CpsC (Supplementary Fig. S5B). Similarly, septal targeting of CpsA-L-FLAG$_3$ and CpsH-L-FLAG$_3$ depends on CpsC, but not on other CPS proteins (Supplementary Fig. S5C). By contrast, deletion of *cpsA*, *cpsC*, *cpsD*, and the GTs *cpsT*, *cpsG*, and *cpsI* dispersed CpsJ-L-FLAG$_3$, indicating that the flippase is recruited late in the cascade (Fig. 3 and Supplementary Fig. S5C). To summarize, the results suggest CpsC recruits CpsA and CpsH, then the GTs, and followed by CpsJ (Fig. 3).

## Recruitment of CpsC to the divisome does not require Pbp2x and Pbp2b transpeptidase activities

If CpsC recruits other CPS enzymes to the divisome, the assembly of the CPS complex may require septal PG synthesis. To test this, we employed methicillin, a β-lactam class antibiotic. Provided that the concentration of methicillin is carefully controlled, it inhibits only the transpeptidase activity of Pbp2x[59,60]. Methicillin-treated cells were harvested right before the onset of cell lysis. We then determined the sites of new PG and CPS synthesis using the pulse-chase experiment described above. While septal PG synthesis was largely abolished by methicillin as judged by the formation of elongated cells (Supplementary Fig. S6A), CPS was still made in regions where FDAA was incorporated (Fig. 4A, Supplementary Fig. S6C). Consistently, methicillin treatment neither affects the positioning of CPS proteins (Fig. 4D) nor the amount and sizes of the CPS produced (Fig. 4B, C, Supplementary Fig. S6B). Deletion of Pbp2b also did not change the location of capsule production (Supplementary Fig. S7). These experiments indicate that although PBP2x is required to recruit CpsC, its transpeptidase activity does not affect the site selection of CPS synthesis.

## CpsC is early in the recruitment cascade of CPS factors

We then sought to determine the order of assembly of the CPS complex. The serotype 2 capsule is produced by the Wzx/Wzy pathway[54–56]. First, the initiating phosphoglycosyltransferase CpsE installs a glucose-1-phosphate moiety on the universal lipid carrier undecaprenyl phosphate (Und-P). Other glycosyltransferases (GTs),

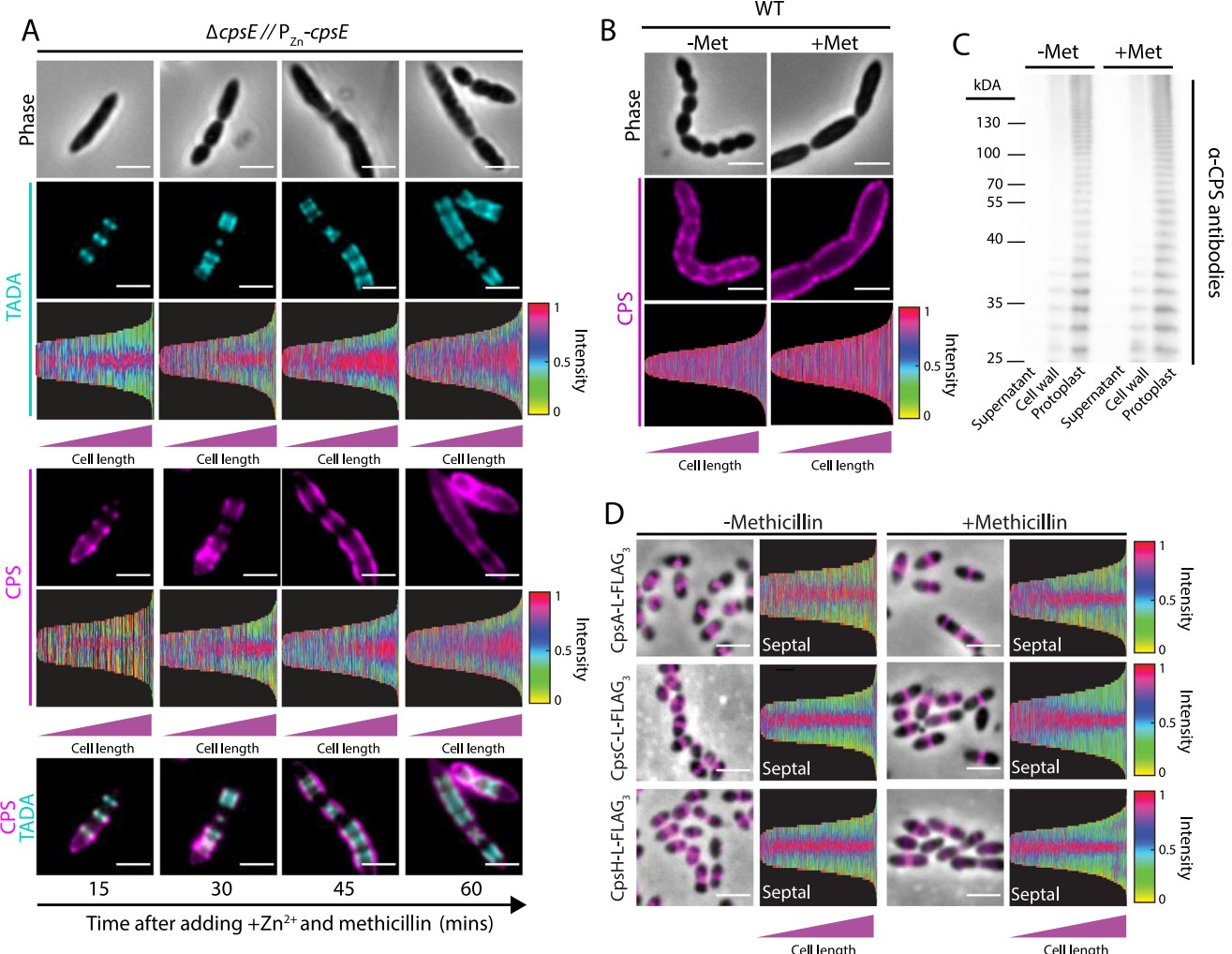

**Fig. 4 | Methicillin treatment does not affect coordination between PG and CPS synthesis. A** Strain NUS0337 [$\Delta cpsE$ $P_{Zn}$-*cpsE*] was grown until it reached the mid-log phase. Cultures were normalized to an $OD_{600}$ of 0.1 and were treated with methicillin for an hour before inducing capsule production by $Zn^{2+}$ supplementation. At the same time, TADA was added to label the site of PG synthesis. Samples were taken every 15 min, heat-killed, and immunostained with anti-serotype 2 capsule antibodies as described in "Methods". Demograph analysis includes data from three biologically independent experiments ($n = 500$). **B** Strain IU1781 (WT) was grown in BHI broth at 37 °C in 5% $CO_2$ to mid-log phase, treated with methicillin for an hour, and immunostained with anti-capsule antibodies. Untreated cells were included here for comparison. Cells were imaged by fluorescence microscopy and shown are the representative phase contrast (top) and epifluorescence (middle) micrographs, as well as the associated demographs (bottom). Demograph analysis includes data from three biologically independent experiments ($n = 500$). **C** Culture of the methicillin-treated cells and untreated control in (**B**) were collected, fractionated, and immunoblotted with anti-capsule antibodies to detect changes in the subcellular localization and the relative amount of CPS. Shown is a representative immunoblot from three biological replicates. Quantification of the CPS amount is provided in Supplementary Fig. S5. **D** Upon methicillin treatment, CpsA, CpsC, and CpsH remained at the septum. Strains NUS0734 [*cpsA-L-FLAG₃* $\Delta cpsE$ // $P_{Zn}$-*cpsE*], NUS0882 [*rpsL1 cpsC-L-sfgfp* $\Delta cpsE$ // $P_{Zn}$-*cpsE*], and NUS3103 [*cpsH-L-FLAG₃* $\Delta cpsC$ $\Delta cpsE$ // $P_{Zn}$-*cpsE*] were treated with methicillin (Met) and immunostained to detect FLAG-tagged proteins as described in (**A**) and the legends of Fig. 1. Bar, 2 μm. Demograph analysis includes data from three biologically independent experiments ($n = 500$).

## Targeting CpsC to the cell pole re-routes CPS synthesis

We wondered if the CPS complex would follow when CpsC is redirected to another part of the cell. To do this, we adopted a recently developed approach called PopZ-Linked Apical Recruitment (POLAR)[61]. POLAR exploits PopZ in *Caulobacter crescentus*, as this protein spontaneously forms foci at the cell poles when overexpressed in other bacterial species[62]. Under this condition, bait proteins fused with the short homo-oligomerization domain of PopZ (i.e., H3H4) will be directed to the pole[61], dragging any interacting partner(s) associated with them (Fig. 5A). To adapt the POLAR system to *S. pneumoniae*, we introduced an N-terminal "i-tag" to stabilize PopZ[63]. The itag-PopZ fusion was then cloned downstream of a strong constitutive promoter ($P_{spxB}$::*itag-popZ*). As a proof-of-concept, we could target itag-mNEONGreen-H3H4 to the cell poles, provided that itag-PopZ was expressed (Supplementary

Fig. S8A). Otherwise, itag-mNEONGreen-H3H4 remained diffused in the cytosol (Supplementary Fig. S8A). Next, we constructed NUS1862 [*cpsC⁺* // $P_{spxB}$-*cpsC-L-sfgfp-H3H4* // $P_{spxB}$-*itag-popZ*] by fusing CpsC-L-sfGFP with a C-terminal H3H4 tag. As expected, CpsC-L-sfGFP-H3H4 formed polar foci in the *popZ⁺* background (Supplementary Fig. S8A), illustrating that the POLAR system robustly targets CpsC to the cell poles.

To determine whether pulling CpsC to the poles changes the sites of CPS synthesis, we labeled the cells with FDAA, harvested them after heat-killing, and stained their capsule with anti-CPS antibodies (Fig. 5A). Expression of PopZ or *cpsC-H3H4* alone did not affect the overall locations of PG and CPS synthesis (Fig. 5C and Supplementary Fig. S8B). However, when CpsC was targeted to the pole (i.e., strain NUS2029 [*cpsC-H3H4* // $P_{spxB}$-*itag-popZ*]), no CPS was detected at the

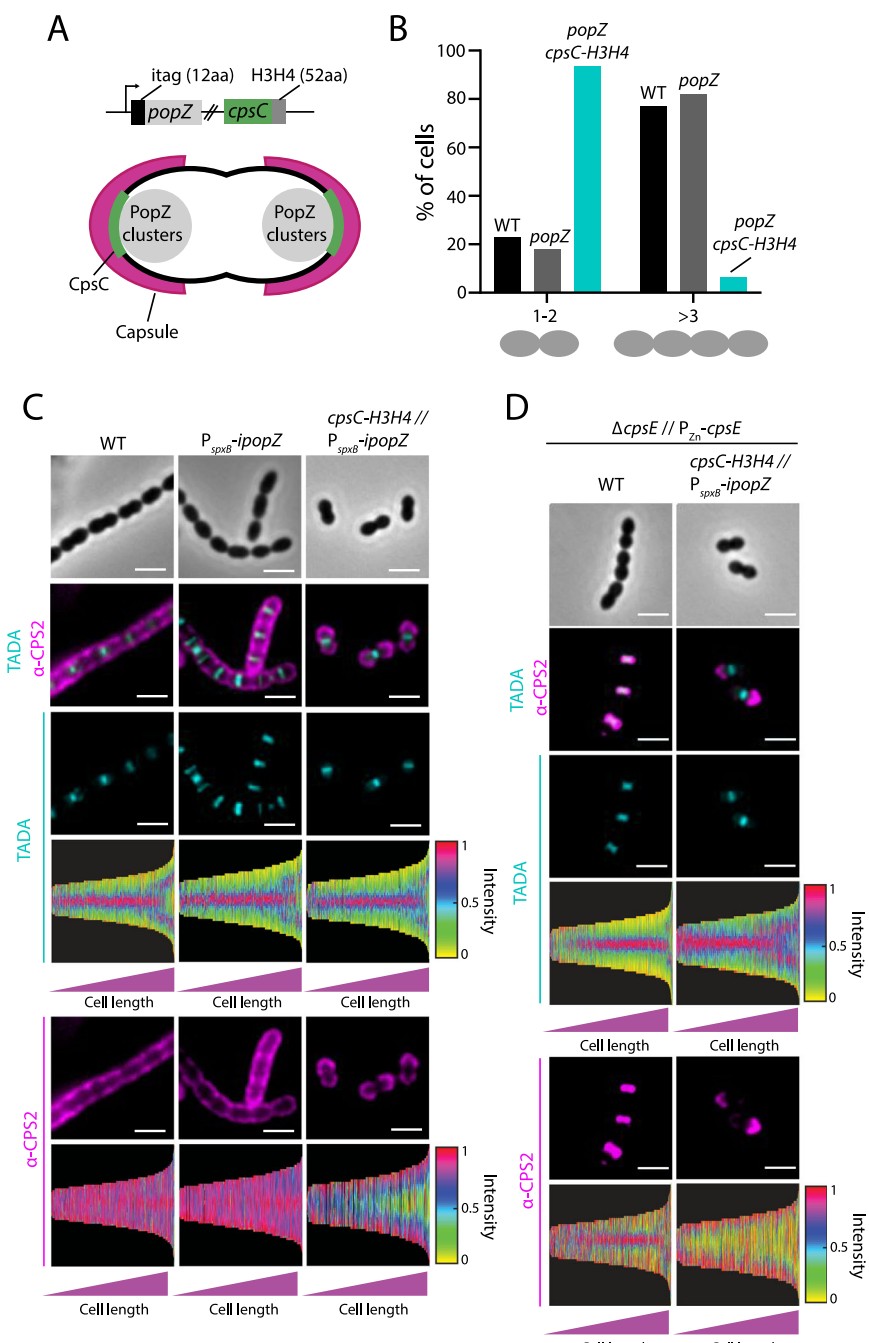

**Fig. 5 | Re-routing CpsC to the cell poles causes polar CPS synthesis. A** Overview of the POLAR system in *S. pneumoniae*. Strain NUS2029 [*cpsC-H3H4* // CEP::P*spxB*-*itag-popZ*] was constructed by fusing PopZ with an i-tag to stabilize it. The fusion protein was cloned downstream of a strong P*spxB* promoter for expression. Under this condition, PopZ spontaneously forms polar foci and pulls CpsC-H3H4 to the cell poles. CPS production will be directed to the poles if CpsC can recruit other components of the CPS complex. **B** Polar CPS synthesis abolished cell chaining. Strains NUS1890 (WT, isogenic parent strain), NUS2335 [P*spxB*-*itag-popZ*] (*popZ*), and NUS2029 [*cpsC-H3H4* // P*spxB*-*itag-popZ*] (*cpsC-H3H4 popZ*) were grown until OD$_{600}$ was between 0.2 and 0.4. Cells were imaged with phase-contrast microscopy and the number of cells per chain was counted and plotted (*n* = 100). **C** Strains NUS1890, NUS2335, and NUS2029 were grown as described in (**B**). Sites of PG and CPS synthesis were determined by FDAA labeling (TADA, teal) for 5 min, heat-killed, followed by immunostaining with anti-CPS antibodies (α-CPS2, pink). Shown are representative micrographs and the corresponding demographs. Demograph analysis includes data from three biological independent experiments (*n* = 500). **D** Strains NUS0267 [*ΔcpsE* // P$_{Zn}$-*cpsE*] and NUS3486 [*cpsC-H3H4 ΔcpsE* // P*spxB*-*itag-popZ* // P$_{Zn}$-*cpsE*] were grown until OD$_{600}$ reached 0.1. CPS production was induced, and new PG was labeled by adding TADA (teal). After 5 min, 1 ml of culture was harvested. The cells were heat-killed, collected by centrifugation, and immunostained with anti-CPS antibodies (α-CPS2, pink). Representative micrographs and the corresponding demographs were shown. Bar, 2 μm. Demograph analysis includes data from three biologically independent experiments (*n* = 500).

septum (Fig. 5C). Instead, CPS was only produced at the poles. Interestingly, the amount of CPS and the polymer length in this strain remained unchanged (Supplementary Fig. S8D and S8E). We also noticed these cells form mostly diplococci (Fig. 5B). Although the cells

were slightly shorter than the wild type (<0.2 μm) (Supplementary Fig. S8C), their overall cell shape and the FDAA labeling pattern remained unaffected (Fig. 5C). Thus, the divisome is unlikely 'back-recruited' to the cell poles by CpsC. Nevertheless, CpsJ-L-FLAG$_3$ was directed to the

pole (Supplementary Fig. S8F). To better track the sites of nascent CPS synthesis, strain NUS3486 [*cpsC-H3H4 ΔcpsE* // P*~spxB~-itag-popZ* // P*~Zn~-cpsE*] was constructed because it enables pulse-chase experiments. Cells were labeled with FDAA as described above, except Zn²⁺ was added to the medium 5 min before heat-killing such that CPS production was briefly induced. Consistent with the capsule being produced at the poles, anti-CPS antibody staining resulted in foci that could be clearly distinguished from the nascent PG synthesized at the septum (Fig. 5D and Supplementary Fig. S9B). Thus, we conclude that the mislocalized capsule complex remained functional and produced CPS outside of the divisome.

### Misplaced CPS exposes vulnerable regions that allow complement deposition

The capsule is critical to evade host complement deposition[6,64,65]. We thus hypothesized that the lack of a capsule at the septal region of the strain expressing CpsC-H3H4 would result in increased complement binding. To test this, we performed a complement deposition assay where the bacteria were incubated with human serum to allow complement binding, followed by double staining using antibodies against CPS serotype 2 and complement C3b, respectively. Fluorescence microscopy showed that wild-type bacteria and bacteria expressing itag-PopZ without the CpsC fusion had a complete CPS surface staining, and only low levels of complement C3b binding were detected (Fig. 6A). As mentioned above, the bacteria that synthesize CPS at the poles as a result of the CpsC-H3H4 fusion did not have capsule at the septal region (Fig. 6A). Instead, C3b could bind to the septal region as can be seen in Fig. 6A, indicating that the lack of capsule at the septal region makes the bacteria vulnerable to complement attacks. To confirm this, we also quantified complement C3b binding by flow cytometry. Results from the flow cytometry showed a clear separation between control (secondary antibody only) and anti-C3b antibody binding in the *cpsC-H3H4* strain (NUS2029) compared to the control strains (Fig. 6B). Together these data show that a lack of capsule at the septal region renders the bacteria susceptible to complement deposition and thus potentially to opsonophagocytosis during infection.

## Discussion

As the outermost layer of the cell, CPS serves as an essential virulence factor for preventing complement deposition and opsonophagocytosis. It also facilitates biofilm formation and mucus escape[54]. The host in turn produces anti-capsule antibodies to attack the CPS layer, which is sufficient to confer immunity against pneumococcal infections caused by the same capsular serotype[55]. Anti-capsule antibody production exerts selection pressure on pneumococci to produce a diverse set of CPSs. Thus, the biochemical armament race between the host and the pneumococcus results in "Red Queen" dynamics, and to date, more than a hundred types of pneumococcal CPS have been identified[54,55]. The diversity of the CPS structure challenges vaccine development. One of the most clinically relevant vaccines for *S. pneumoniae* is the pneumococcal conjugate vaccine PCV13. Yet, it covers only ~60% of the circulating serotypes in some Asian countries[66,67]. Therefore, an updated vaccine is needed. Understanding the mechanism of CPS synthesis promises to find new avenues of pneumococcal vaccine development.

Most of the CPSs are produced by the Wzx/Wzy-dependent mechanism[54], they often share the same lipid carrier with PG[68]. CPS made by this mechanism is also covalently attached to PG[16,68]. Besides controlling the length of the CPS polymer, the CpsCD complex governs the location of CPS synthesis[10,11]. As shown by the pulse-chase experiments in this study, CpsC, CPS synthesis, and PG synthesis are co-localized at midcell. CpsC is recruited to the divisome possibly by interacting with a late cell division protein. Subsequently, CpsC triggers a recruitment cascade that brings CpsA, CpsH, the GTs, and finally the CpsJ flippase to the divisome. Since we were unable to determine

the subcellular localization of CpsE, CpsP, and CpsK, we are unsure whether they are recruited to the same complex or remained diffused. As their disruptions did not affect the location of CpsC, CpsA, CpsH, and CpsJ, they are likely downstream of CpsJ if they are indeed part of the CPS-divisome complex (Fig. 3). Intriguingly, CpsA and CpsC are conserved in most serotypes, whereas CpsJ, CpsH, and other GTs are not[54,56]. If the same recruitment cascade is employed in other serotypes, how can CpsA and CpsC interact with these highly variable CPS factors? We propose the GTs may form a subcomplex before interacting with CpsA. The GT-CpsA complex then recruits the cognate CpsJ. Provided that the GT subcomplex co-evolves with CpsA and CpsJ, and CpsH maintains its interaction with CpsC, the recruitment mechanism will remain functional. Consistently, the amino acid sequence of CpsJ strongly correlates with the cargo it transports[69]. This observation suggests the GTs and CpsJ may co-evolve because the CPS precursors likely track with the GTs that produce them. The polymerase co-polymerase 2b domain of CpsC may also bridge its interaction with CpsH[9,16]. Ongoing experiments are underway to examine these hypotheses.

The elongasome is responsible for producing a substantial amount of PG. If the CPS complex only associates with the divisome, how can the entire cell be covered with CPS? Unlike rod-shaped bacteria, the elongasome of ovococci remains at the septum throughout the cell cycle[28,32] (Fig. 7A). Just before cell division, the elongasome and the divisome arrive at the septum simultaneously, forming the pre-septal ring[17] (Fig. 7A). Once cytokinesis is initiated, the pre-septal ring is activated and produces PG. At the same time, the two complexes rotate inward towards the septal annulus, where the divisome migrates faster than the elongasome (Fig. 7B). The elongasome contains the tentative space-making enzyme PcsB and its regulator FtsEX[32]. According to a model proposed by the laboratory of Cecile Morlot[46], this space-making enzyme splits the septal PG to make room for the newly synthesized peripheral PG to incorporate. As a result, the mature PG is a hybrid of peripheral and septal PG, a product of both complexes[46]. This remodeling process likely expands and fortifies the sacculi by the concerted action of turgid pressure, PG hydrolysis, and synthesis. Thus, once the CPS-containing septal PG reaches the elongasome ring, it would be distributed to the rest of the cell (Fig. 7B, C).

Remarkably, re-routing CpsC to the cell poles causes apical capsule synthesis. This observation provided several interesting insights. First, PG at the cell pole is thought to be stable for many generations[70]. Since the CPS complex remains functional at the cell pole, it implies that unlike WTA[71], CPS can be installed on mature PG. Consistently, disruption of septal PG synthesis by methicillin treatment did not affect CPS synthesis (Supplementary Fig. S6). How the long CPS chains can be transferred to the mature PG warrants further investigation. While we did not exclude the possibility that there are other cell surface anchors for CPS, the non-diffusive signal of the capsule and the lack of growth phenotype suggest CPS has been transferred to the PG and is no longer attached to Und-P (Fig. 5 and Supplementary Fig. S9). In addition, immunoblotting experiments showed no difference in the subcellular location, amount, and chain length of the CPS produced even though the CPS complex is spatially uncoupled from the divisome (Supplementary Fig. S8). As there is no evidence of divisome back-recruitment, these results hardly fit the model whereby the rate of CPS synthesis is allosterically controlled by cell division proteins. Instead, we propose that the recruitment of the capsule complex to the septum is important for protecting against complement deposition (Fig. 6) and for chain formation (Fig. 5). Chaining is known to promote competence[72], which is important for pneumococci to acquire antibiotic resistance[73]. Cells of strain NUS2029 [*cpsC-H3H4* // P*~spxB~-itag-popZ*] are diplococci despite being fully encapsulated. This suggests that the capsule must be placed at the correct location to facilitate chaining. Since the strain lacks CPS at midcell, the septum is exposed to complement and

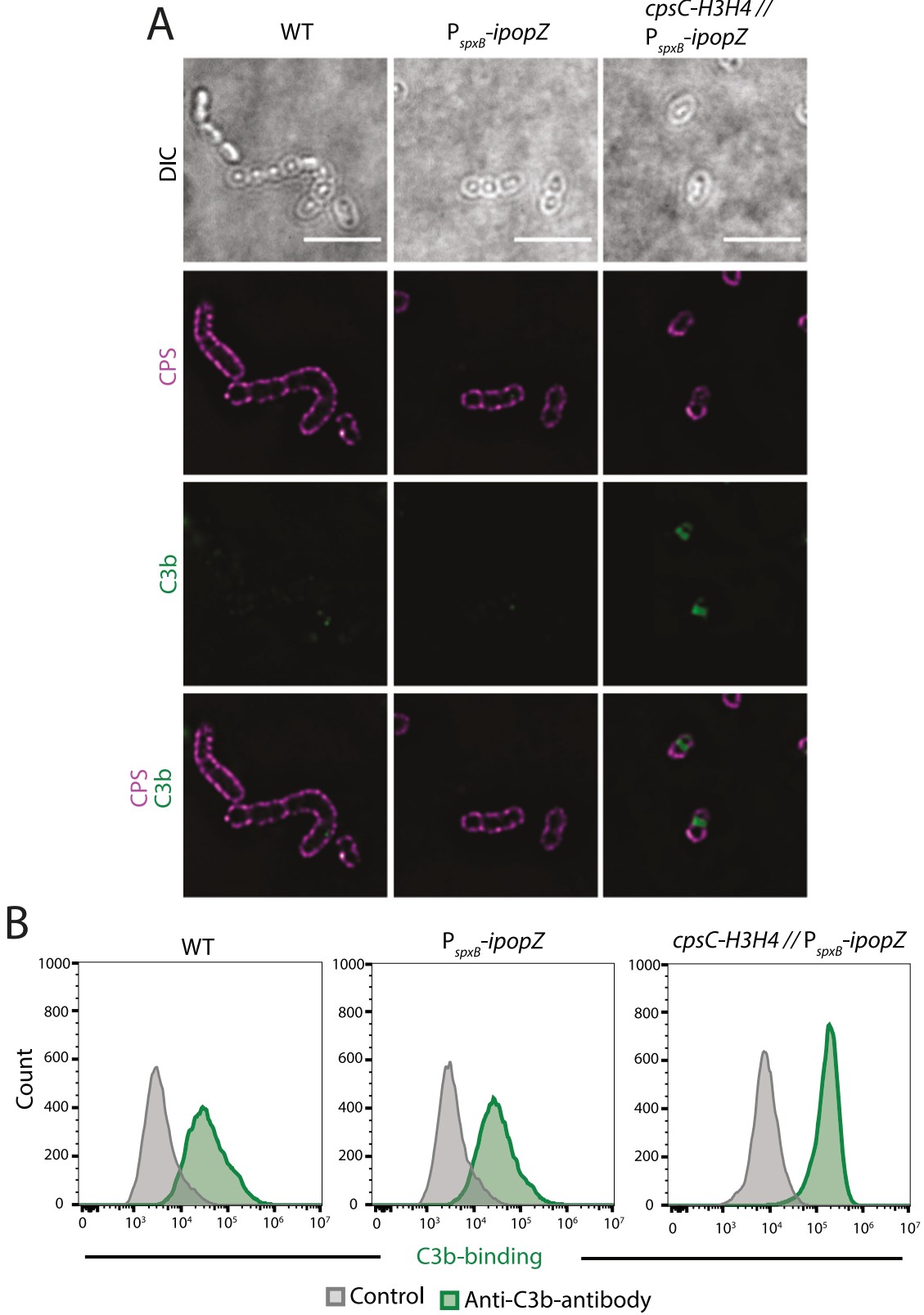

**Fig. 6 | Lack of capsule results in increased complement C3b binding at the septal region. A** Strains IU1781 (wild-type; WT), NUS2335 (P$_{spxB}$-*ipopZ*), and NUS2029 (*cpsC-H3H4* // P$_{spxB}$-*ipopZ*) were grown until an OD$_{620}$ of 0.2. After heat-killing and human serum incubation, CPS and capsule detection were done by immunostaining using anti-CPS antibodies and anti-C3b antibodies. Representative fluorescence microscopy micrographs of CPS (red), complement C3b (green), and overlay of CPS and complement C3b are shown. Scale bar = 5 μm. **B** Representative histograms of complement C3b binding to strains IU1781, NUS2335, and NUS2029 using flow cytometry. The strains were grown, heat-killed, and incubated with human serum. Immunostaining was done by using anti-C3b antibodies. Bacteria from each strain, incubated without anti-C3b antibodies were used as controls. All biologically independent experiments were performed three times with similar results.

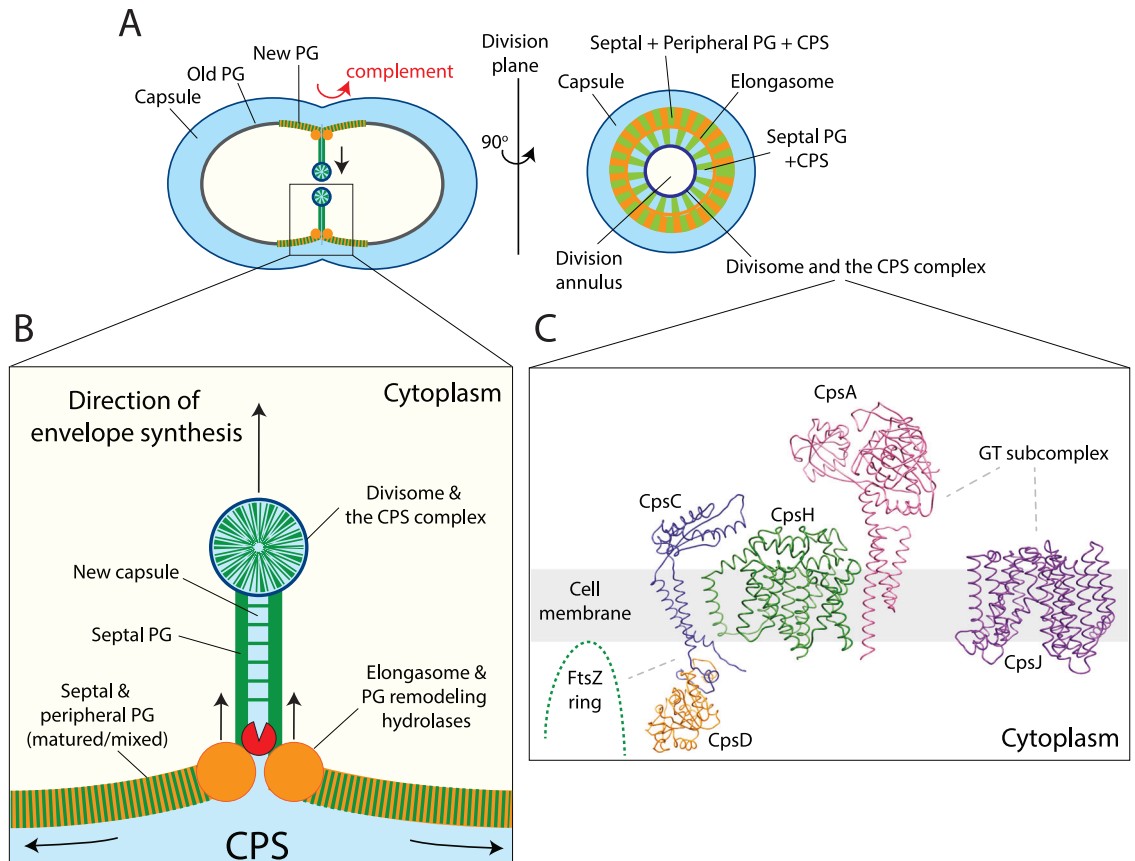

**Fig. 7 | Interplay between PG synthesis, cell division, and CPS synthesis. A** CPS synthesis is closely integrated with PG synthesis to avoid attacks by complement (red arrow) or other host immune factors. We propose the nascent septal PG (green) is already covered by CPS (light blue) before reaching the cell surface. The encapsulated septal PG (green and blue) is remodeled by the elongasome (orange) to synthesize the matured PG (green, orange, and blue), thereby spreading CPS to other parts of the cell. In this model, the mature PG contains products of the divisome-CPS complex and the elongasome. **B** The septal PG/CPS made by the divisome-CPS complex is presumably cleaved by a PG remodeling enzyme, likely the PcsB-FtsEX complex. This process will split the septum, forming new sidewalls of the daughter cells. As the turgor pressure pushes the leading edge of PG hydrolysis, the sidewall will be 'elongated' by a concerted action of the PG synthases (class A PBPs and RodA-Pbp2b). **C** The CPS complex is directed to the divisome, starting with the CpsCD tyrosine kinase. CpsCD then recruits CpsA and CpsH, followed by the GT subcomplex, and lastly CpsJ. As the locations of other CPS factors remained unclear, they are not shown in this figure. The structural models of CpsCD, CpsH, CpsH, and CpsJ illustrated here were obtained by AlphaFold and AlphaFold2 Advance[79,80].

perhaps antibody attacks. As a result, while polar capsule synthesis did not cause any noticeable defects in laboratory conditions, it likely affects survival inside the host.

## Methods

### Bacterial strains and growth conditions

A list of strains used in this study is provided in Table S1. Unless otherwise specified, *S. pneumoniae* was grown in brain heart infusion (BHI) broth (Thermo Fisher, CM1135B) or tryptic soy agar (TSA)-II plates supplemented with 5% defibrillated sheep blood (BD, 221261) at 37 °C in 5% $CO_2$. Antibiotics were purchased from Sigma-Aldrich and used at a concentration of 0.3 μg/mL for erythromycin, 250 μg/mL for kanamycin (Kan), and 300 μg/mL for streptomycin (Str), respectively. *S. pneumoniae* was transformed after inducing natural competence with PCR amplicons or Gibson assembly products[74,75]. PCR fragments were prepared using Phusion High-Fidelity DNA Polymerase (NEB, M0530S), diagnosed by agarose gel electrophoresis, and purified using the QIAquick PCR Purification Kit (Qiagen, 28106). Oligonucleotides for synthesizing PCR amplicons are listed in Table S2. The amino acid sequence of the FLAG epitope is DYKDDDDK. FLAG3 indicates three 5 tandem FLAG epitopes. "L" is a linker (GSAGSAAGSG). Allelic exchanges were done by positive and negative selection using the Sweet Janus[76] or the Janus cassettes[77]. Transformants were plated on blood agar supplemented with the appropriate antibiotics and streak-purified. The genetic constructs in the transformants were validated by PCR using GoTaq DNA polymerase (Promega, M712) and confirmed by Sanger sequencing.

To measure growth, overnight cultures of the indicated strains were diluted in BHI broth to a starting optical density at 600 nm ($OD_{600}$) of 0.01. The diluted cultures were grown at 37 °C in 5% $CO_2$, and the $OD_{600}$ readings were measured every 30 min using the GeneSys 30 spectrophotometer (Thermo). For strains that require $Zn^{2+}$ to grow, cultures were supplemented with $ZnCl_2$ and $MnCl_2$ at a final concentration of 0.5 mM and 50 mM, respectively. To remove $Zn^{2+}$ for protein depletion, cells were collected by centrifugation at $16,100 \times g$ for 5 min at room temperature, washed once with an equal volume of BHI, and resuspended in BHI broth with or without $ZnCl_2$ and $MnCl_2$. Cells were harvested for live imaging and immunofluorescence microscopy at the appropriate time points.

### Immunofluorescence microscopy

Microscope slides were marked with a Liquid Blocker Super Pap Pen (EMS; 71310) to trap liquid. Slides were coated by adding 100 μl of 0.01% poly-L-lysine solution (Merck P4707) and incubated overnight at 4 °C. Excess poly-L-lysine was removed by pipetting, air-dried, and used immediately. Cells of the indicated strains were grown in BHI broth at 37 °C in 5% $CO_2$ to an $OD_{600}$ of 0.4. Cells from 1 ml of cultures were harvested by centrifugation at $16,100 \times g$ for 5 min at room

temperature, washed once with 1 ml of 1x PBS (137 mM NaCl, 2.7 mM KCl, 10 mM $Na_2HPO_4$, 1.8 mM $KH_2PO_4$, pH 7.4), and resuspended in 1 ml of 4% (w/v) paraformaldehyde (Electron Microscopy Sciences, 157-4). The mixture was incubated for 15 min at room temperature and 45 min on ice. Fixed cells were collected by centrifugation at $16,100 \times g$ for 5 min at room temperature. Pellets were washed once with 1x PBS and resuspended in 100 µl of cold GTE buffer (50 mM glucose, 20 mM Tris-HCl pH 7.5, 1 mM EDTA). Half of the cell suspensions (50 µl) were added to poly-L-lysine-coated slides and incubated overnight at 4 °C. The suspension was removed, and the slides were washed by adding PBS Triton (1x PBS with 0.2% (v/v) Triton X-100) and incubated for 5 min at room temperature. Next, the PBS Triton solution was discarded, and the slides were immersed in cold methanol and kept at −20 °C for 10 min. After air-dried, 50 µl of blocking buffer (1x PBS with 5% (w/v) skimmed milk) was added to the slides and incubated for 1 h at room temperature. Slides were washed twice with PBS Triton, incubated with anti-FLAG polyclonal antibody (Sigma F7425) at 1:200 dilution for 1 h at room temperature, washed twice with PBS Triton, and incubated with secondary anti-rabbit-Alexa 647 (Thermo scientific A-31573) at 1:100 dilution for 1 h at room temperature. Both antibodies were diluted in the blocking buffer. Lastly, the slides were washed twice with PBS Triton and air-dried. To mount them, 5 µl of Slowfade Gold antifade reagent (Invitrogen; S36936) was added before placing the coverslips.

### Pulse-chase experiments
Cells of the indicated strains were grown to an $OD_{600}$ of 0.1 to 0.3. Cultures were diluted to an $OD_{600}$ of 0.1, then methicillin was added to a final concentration of 0.125 µg/ml and incubated at 37 °C in 5% $CO_2$ for 1 h. HADA (3-[7hydroxycoumarin]-carboxamide-D-alanine), TADA (TAMRA 3-amino-D-alanine), $ZnCl_2$, and $MnCl_2$ was added to a final concentration of 125 µM, 125 µM, 0.5 mM, and 50 mM, respectively. The cultures were incubated at 37 °C in 5% $CO_2$ and samples were taken every 15 min for heat inactivation at 65 °C for 45 min. Cells were harvested by centrifugation at $20,000 \times g$ for 2 min at room temperature and washed once in 1x PBS. Then, 100 µl cross-absorbed anti-serotype 2 CPS antiserum (SSI) was added at a 1:400 dilution, and the mixture was incubated on ice for 5 min. Cells were collected by centrifugation, washed once with 1x PBS, and resuspended in 100 µl PBS. CPS was detected by adding goat anti-rabbit IgG conjugated with Alexa Fluor 647 (Thermo Fisher, A31573) at a 1:100 dilution and incubated on ice for 5 min. Labeled cells were washed with 1x PBS, dispersed in 5 ml of mounting medium (Thermo Fisher, S36936), and visualized using an IX81 microscope. Demographs were made with Oufti[78] using the demograph function under the tools and signal statistics tab. The parameters are: 'areaMin' = 150, 'areaMax' = 15,000, 'cellwidth' = 20, 'wspringconst' = 0.2, and 'split_threshold' = 0.43. More than 500 cells were analyzed for each demograph. 'maximum number of cells to be included in final demograph' was set to 500 and cells were randomly selected by the software 'randomN'. To measure the area of new PG, CPS, and area of the cell with MicrobeJ, the "Bacteria" and "Maxima" functions were used. Under the "Bacteria" tab, "Medial Axis" was selected, and the thresholding function was adjusted accordingly to accurately detect the bacterial cells. To eliminate the detection of debris, "Area [µm²]" was set to "0.15-max." The options "Exclude On Edges," "Shape descriptors," "Chain," and "Segmentation" were checked. "Foci" was selected to detect fluorescent signals. "Tolerance" and "Z-score" were adjusted accordingly to optimize the detection. Under the "association" tab, "Parent: Bacteria" and "Location" and "Outside" for CPS and "Inside" for PG were selected.

### Immunoblotting
Cells of the indicated strains were grown in BHI to an $OD_{600}$ of 0.2. Where indicated, $ZnCl_2$ and $MnCl_2$ were added to the culture to induce capsule production at 37 °C in 5% $CO_2$ for 1 h. Cultures were normalized to an $OD_{600}$ of 0.2, and 1 ml of culture was centrifuged at

$16,100 \times g$ for 1 min at room temperature. Biochemical fractionation was done where indicated. Briefly, the spent medium was collected, and the cell pellets were washed once with 1x PBS. Samples were resuspended in 200 µl of protoplast buffer (50 mM of Tris-HCl pH 7, 50 mM $MgCl_2$, and 20% (w/v) sucrose) for biochemical fractionation. Peptidoglycan was digested by adding 40 units of mutanolysin (Sigma M9901) and 25 µg of lysozyme (Axil Scientific L-040-25) and incubated overnight at room temperature. Peptidoglycan released was separated from the protoplasts by centrifugation at $3000 \times g$ for 10 min at room temperature. Cells were lysed by resuspending in an equal volume of 1x PBS. Samples were mixed with an equal volume of 2x Laemmli buffer. If the blot is used for measuring the amount and sizes of the CPS, 2 µl of proteinase K (Qiagen 19133) was added and the sample was incubated at 56 °C for 1 h to degrade cellular proteins. Samples were resolved on a 10% (w/v) SDS-PAGE gel. The gel was transferred to a PVDF membrane and incubated for 30 min at room temperature in the blocking buffer (1x PBS with 0.05% (v/v) Tween 20 and 5% (w/v) skimmed milk). The membrane was probed with anti-serotype 2 CPS antiserum (SSI 16745) at a 1:5000 dilution, anti-FLAG antibodies (Sigma F7425) at a 1:2500 dilution, or anti-GFP antibodies at a 1:2500 dilution (Abcam ab1218) in blocking buffer at 4 °C overnight. The blot was washed twice with PBST (PBS with 0.05% (v/v) Tween 20) and detected by goat anti-rabbit HRP antibodies (Thermo Fisher A16110) or goat anti-mouse HRP antibodies (Thermo Fisher G-21040) at a 1:10,000 dilution. The membranes were washed three times with PBST before detection with a chemiluminescent substrate (Thermo Fisher 34580).

### Enzyme-linked immunosorbent assay (ELISA)
Strain HMS0002 [$rpsL1$ $\Delta cpsE$] was grown in 50 ml of BHI broth until the $OD_{600}$ was between 0.6 and 0.8. Cells were washed once with 1 ml of 1x PBS and inactivated by incubating at 65 °C for 45 min. Heat-killed cells were resuspended in 1x PBS at a final $OD_{600}$ of 0.9. For cross-adsorption, 8 µl of anti-serotype 2 CPS antiserum was mixed with 280 µl of heat-killed cells and 8 ml of 1x PBS with 1% (w/v) bovine serum albumin (BSA). The mixture was incubated overnight at 4 °C and the heat-killed cells were removed by centrifugation and filtered through a 0.22 µm filter (Costar CLS8160). The adsorbed antiserum was stored at 4 °C until use.

Cells of strains IU1781 (isogenic parent strain) and NUS2029 [$cpsC$-$H3H4$ // P$_{spxB}$-$itag$-$popZ$] were grown in BHI to a final $OD_{600}$ of 0.2. Cultures were normalized to an $OD_{600}$ of 0.4, and 1 ml of culture was centrifuged at $16,100 \times g$ for 1 min at room temperature. Pellets were resuspended in 100 µl of 1x PBS. Samples were treated by adding 4 µl of proteinase K (Qiagen 19133) and incubated at 56 °C for 1 h. The supernatant fraction was removed by centrifugation and pellets were resuspended in an equal volume of 50 mM carbonate/bicarbonate buffer (Sigma C3041). The suspension was added to the ELISA plates (Thermo Fisher 442404 and incubated overnight at 4 °C. ELISA plates were washed with 1x PBS and blocked by incubating in 1x PBS with 1% (w/v) BSA for 1 h at room temperature. Wells were washed four times with PBST, and cross-adsorbed anti-serotype 2 CPS antiserum (SSI Diagnostica) was added at a 1:1000 dilution, followed by incubation for 1 h at room temperature. Plates were washed four times with PBST and goat anti-rabbit HRP antibodies (Thermo Fisher A16110) were added at a 1:1000 dilution in 1X PBS with 1% (w/v) BSA. After washing four times with PBST, antibodies were detected by a colorimetric OPD (o-phenylenediamine dihydrochloride) substrate (Sigma, P9187-50SET) according to the manufacturer's protocol. Optical densities at 450 nm were recorded with a Tecan plate reader.

### Complement deposition assay
Bacteria were grown until $OD_{620}$ of 0.2 at 37 °C in 5% $CO_2$. When the cultures reached $OD_{620}$ of 0.2, 2 ml cultures were heat-killed at 65 °C for 1 h followed by washing in 1 ml 1x PBS. Bacteria were then incubated in 300 µl of 20% normal human serum (NHS) (Sigma S1-100 ml) at room

temperature for 30 min and washed in 1 ml of 1x PBS. For antibody staining, bacteria were first incubated in 300 μl of rabbit anti-capsule serotype 2 antibodies (SSI 16745) in a dilution of 1:150 at room temperature for 30 min and in 300 μl of goat anti-C3b antibody (Calbiochem 204869) in a dilution of 1:150 at room temperature for 30 min. Each staining was followed by washes in 1 ml of 1x PBS. Subsequently, the bacteria were incubated in 300 μl of AlexaFluor 594 conjugated donkey α-rabbit IgG (Invitrogen A-21207) at room temperature for 30 min and in 300 μl of AlexaFluor 488 conjugated donkey α-goat antibody (Invitrogen A-11055) at room temperature for 30 min. Each incubation was followed by washes with 1 ml of 1x PBS. Bacteria were then fixed by 100 μl of 4% PFA at room temperature for 10 min, washed in 1 ml of 1x PBS, resuspended in 40 μl of 1x PBS and 5 μl was spread onto cover slides. The cover slides were mounted by Vectashield anti-fade mounting medium (Vector Laboratories). Fluorescence microscopy was performed using a Deltavision Elite microscope (Applied Precision) and image analysis was done using ImageJ. For flow cytometry analysis, samples were grown, heat-killed, incubated with 20% N-hydroxysuccinimide NHS, and stained with anti-C3b antibody and secondary antibody as described above. Bacteria were fixed in 100 μl of 4% PFA at room temperature for 10 min, washed in 1 ml of 1x PBS, and resuspended in 300 μl of 1x PBS. The acquisition was done using a Gallios flow cytometer (Beckman Coulter) and data analysis using FlowJo (v10.8.1).

### Reporting summary

Further information on research design is available in the Nature Portfolio Reporting Summary linked to this article.

## Data availability

All data generated or analyzed during this study are included in this published article (and its supplementary information files). Source data are provided with this paper.

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

## Acknowledgements

We thank Ms. Jade Ye-Yu Chun and other members of the L.T.S. laboratory for the useful discussions. This work is supported by grants from the National University of Singapore Start-up grant (NUHSRO/2017/070/SU/01), the National Research Foundation Fellowship (NRFF11-2019-0005), the Ministry of Education Academic Research Tier 2 Fund (MOE-T2EP30220-0012), the Swedish Research Council, the Knut and Alice Wallenberg foundation, the Swedish Foundation for strategic research (SSF), the Torsten Söderberg Foundation, the Stockholm county council, and the Clas Groschinsky Foundation.

## Author contributions

R.N. and L.-T.S. designed research; R.N., S.B., and K.B. performed research; R.N., S.B., K.B., S.N., and B.H.-N. analyzed data; R.N. and L.-T.S. wrote the paper; S.B., K.B., S.N., and B.H.-N. contributed to the manuscript. All authors read and commented on the manuscript.

## Competing interests

The authors declare no competing interests.
