## [Peer Review File · Nature Communications]

The divisome but not the elongasome organizes capsule synthesis in *Streptococcus pneumoniae*.Reviewer #1 (Remarks to the Author):

The manuscript submitted by Nakamoto and colleagues is a refreshing and important manuscript that relates two processes crucial for bacteria – the synthesis of peptidoglycan and the synthesis of the surface capsular polysaccharide.

Essentially, authors propose that CpsC, an enzyme involved in the synthesis of the pneumococcal capsule, recruits the capsule synthetic machinery to the subcellular region where the capsule must be produced.

Authors not only confirmed previous observations related to the synthesis of CPS that had been made by other groups, but they integrated this knowledge into a model proposing that CpsC determines the localization of the capsule synthesis machinery.

Authors started by enquiring whether the localization of CpsC is dependent on proteins involved in the assembly of the division septum or on other proteins of the CPS operon.

Having settled that CpsC co-localizes with the divisome, and not with the elongasome, authors determined the order of the assembly of the capsule synthetic machinery, specifically, that CPS complex assembly starts with CpsCD, then CpsA and CpsH, the glycosyltransferases, and finally CpsJ.

Finally, authors claim the CpsCD is sufficient to direct the capsule synthesis as they were able to target CpsC to the cell pole and, consequently, reposition CPS synthesis in a way that is independent on the PGN synthesis.

Moreover, author confirmed that the “inability to synthesize CPS at the septum sensitizes the cell towards complement deposition, indicating that the location of the CPS complex could be important for virulence”.

I believe that this work is of great interest to those interested in understanding how bacteria assemble their cell surface in a way that keeps them protected from the infected host. Authors present evidence that supports their conclusions but should consider strengthening certain parts of their manuscript (see below).

Major points

Point A.

Lines 91-92. Authors constructed a mutant that expresses CpsC-L-sfgfp encoded at the native locus and stated that “cpsC-L-sfgfp remained functional because it complemented cpsC (Supplemental Fig. 1A)”.

Could authors also include here the strain Delta cpsC // Delta cpsE // Pzn-cpsE so that the reader can observe, and compare, the effect exerted by the absence of a functional CpsC (or the presence of a non-functional one)?

I believe this is important as different groups have reported that the presence of a non-functional CpsC in encapsulated pneumococcal mutant strains may result in bacteria with capsule at the poles – a phenotype that authors have reported in the manuscript when they tried to direct capsule synthesis to the bacteria poles.

Point B.

Lines 92-93. Author state that “...CPS production was restored to a normal level when Zn²⁺ was added to the medium (Supplementary Fig. 1B)”.

It is very difficult to confirm this statement as there are non-encapsulated bacteria in both strains shown in panels of Sup. Fig 1 after 60 min of induction with zinc.

Could authors use an additional way to demonstrate that the CPS production/full encapsulation is restored to normal levels when CpsC-L-sfGFP is produced instead of CpsC (authors could

determine the percentage of fully encapsulated bacteria or compare the amount of CPS produced in equivalent number of bacteria).

It would also be important to demonstrate that the presence of zinc does not interfere with the production of capsule in the parental bacteria as the presence of different environmental cues may result in bacteria altered CPS production.

Point C.

Line 96. "...confirmed that CPS synthesis, PG synthesis, and CpsC are co-localized (Fig. 1)."

It is difficult to analyze this figure 1 as it seems that the same color is being used to label CPS in the microscope image and in the demograph. This doesn't happen in the demographs associated with the localization of CpsC-L-sfgfp or the recently synthesized PG.

It would be helpful if authors used the same color code in all demographs in the manuscript and include the color/localization code in the figure.

Is it possible to associate a septal localization that seems to be observed in the demograph to a number? It could be the percentage of cells with mid cell localization. This would help to analyze data presented in demographs later in the manuscript (see below).

Additional information may be extracted from the demographs, namely by further analysing the patterns in short/younger? and longer/old? cells. It seems that PG incorporation precedes cpsC-L-sfGFP localization, which precedes CPS incorporation (which is in accordance with authors' claims).

Point D.

Line 110-122. Authors have constructed mutants that lack different enzymes of the divisome and that express a fluorescently labeled CpsC. The observation of a signal scattered throughout the cell membrane was interpreted as delocalization.

Do authors know if the levels of delocalized CpsC-L-GFP are similar in the different mutants? If one of the mutations would lead to an increase in the amount of CpsC, then what it seems to be a delocalization could in fact be overexpression.

Authors could use the same procedure that they have used to show that CpsC-L-GFP is not cleaved (Figure S1C).

Point E.

Lines 157-159. "CpsC-L-FLAG3 was fully functional with minimal proteolysis (Supplemental Fig. 4), and it was localized to the septum like the sfGFP-tagged CpsC (Figs. 3, Supplemental Fig 3) "

Can authors show that the expression of Cps-L-FLAG3 results in bacteria that are fully encapsulated in the presence of zinc (to induce the expression of CpsE)?

Point F.

Lines 201-203 "... (Supplemental Fig. 6A), CPS was still made in regions where FDAA was incorporated (Fig. 4A, Supplemental Fig. 6C). Consistently, methicillin treatment neither affects the positioning of CPS proteins (Fig. 4D) nor the amount and sizes of the CPS produced"

Could authors include some information relatively to the incorporation of capsule in the absence of methicillin? Do they have a recovery of fully encapsulated bacteria?

Point G.

Lines 205-207 "These experiments indicate that although PBP2x is required to recruit CpsC, its transpeptidase activity does not affect the site selection of CPS synthesis."

How do authors know if pbp2x is inactivated?

Point H.

Lines 220-222 "As expected, CpsC-L-sfGFP-H3H4 formed polar foci in the popZ+ background (Supplemental Fig. S8A), illustrating that the POLAR system robustly targets CpsC to the cell poles."

Authors should present data from cpsC-H3H4 without ipopZ. If they are correct, and if cpsC-h3h4 is functional, bacteria should be fully encapsulated.

Point I.

Lines 237-240 "...with the capsule being produced at the poles, anti-CPS antibody staining resulted in foci that could be clearly distinguished from the nascent PG synthesized at the septum (Fig. 5D and supplemental Fig. 9B). Thus, we conclude that the mislocalized capsule complex remained

functional and produced CPS outside of the divisome. "

To strengthen the claim that mislocalized CpsC is recruiting a functional capsule complex to places outside the divisome, authors should show that one of the other CPS proteins has also a polar localization in these strains expressing CpsC-L-H3H4 in the presence of ipopZ.

Minor points

m1.

Line 138-139. Figure 2I.

Is the WT in this figure the delta kphA mutant? To prevent misinterpretations, it may be useful to state clearly in the figure that one of the mutants is delta kphA delta mreC while the other is, I assume, delta kphA

m2.

Lines 180-181. "As expected, CpsA-L-FLAG3, CpsJ-L-FLAG3, and CpsH-L-FLAG3 were intact and localized at the septum (Supplemental Fig. 4A and 5B)."

Do authors know if these constructs are functional? This information should be clearly stated.

m3.

Lines 184-187. "We propose that CpsC-L-FLAG3 arrives at the divisome first, because Δ cpsC delocalized CpsA, CpsJ, and CpsH (Fig. 3, Supplemental Fig. 5B). Furthermore, CpsC localization is independent of other CPS factors, except CpsD, which was shown to stabilize CpsC (Fig. 3; Supplemental Fig. 5A)."

It is difficult to associate a demograph to a conclusion of "localized vs non-localized". It gets even more difficult as in Figure 3 there is no legend stating what is the meaning of different colors. Could authors include in each demograph whether the FLAG-labeled protein has a septal localization?

Authors should take a particular look at Figure S5 panel A. One can observe in the microscope image that the deletion of CpsD interferes with septal localization of CpsC-L-FLAG3. However, the demograph seems like others where CpsC-L-FLAG3 has a clear septal localization.

It would help the readers, if authors would associate a code to each demograph of Fig 3 where they would identify if the labeled protein has the expected septal localization.

In Figure 3 and S5, were bacteria expressing capsule? I am assuming that these observations were done in the absence of zinc but it is not clear. Authors should state clearly that that these localizations were carried out in the absence of capsule (as bacteria were propagated in BHI without zinc)

m4.

Lines 220-221. "As expected, CpsC-L-sfGFP-H3H4 formed polar foci in the popZ+ background (Supplemental Fig. S8A), illustrating that the POLAR system robustly targets CpsC to the cell poles.

Could authors strengthen this claim by presenting percentage of cells with CpsC at the poles in the different conditions?

m5.

Line 687 "were then immunostained with anti-FLAG and anti-serotype 2 capsule antibodies as described in..."

there is no capsule staining in this figure. Could authors correct the figure legend or include the reported data in the figure.

m6.

Line 690. "Shown are the demographs of the fluorescent channels depicting the location of the FLAG tag fusion proteins as well as the tentative recruitment pathway. Representative micrographs are presented in Supplemental Fig. 4. "

Could authors confirm this sentence? I believe they were referring to Sup. Fig. 5.

m7.

Lines 227-228. "However, when CpsC was targeted to the pole (i.e., strain NUS2029 [cpsC-H3H4 // PspxB-itag- popZ]), no CPS was detected at the septum (Fig. 5C). Instead, CPS was only produced at the poles. "

Is CpsC-H3H4 functional?

m8.

Lines 249-251 "As mentioned above, the bacteria that synthesize CPS at the poles as a result of the CpsC-H3H4 fusion did not have capsule at the septal region (Fig. 6A). Instead, C3b could bind to the septal region as can be seen in Fig. 6A, "

Could authors use other color to label CPS? In a dark background it is very difficult to see red signals. It seems that the bacteria cpsC-H3H4 // PspxB-ipopZ are fully encapsulated.

m9

Line 667 and in other locations of the manuscript.

Fluorescence microscopy instead of fluorescent microscopy?

Reviewer #2 (Remarks to the Author):

The work submitted by Nakamoto et al "The divisome but not the elongasome organizes capsule synthesis in *Streptococcus pneumoniae*" studies the spatiotemporal orchestration of capsule synthesis with peptidoglycan biosynthesis and cell division. The authors showed that CPS synthesis initiates from the division septum and that assembly of the synthesis complex is organized by the bacterial tyrosine kinase system CpsCD. Recruitment of the proteins to the division septum is associated with the divisome, but not the elongasome. Using the POLAR system, CPS synthesis can be repositioned to the cell pole resulting in diplococci that lack CPS at the septum. Nakamoto et al propose the importance of septal CPS for chain formation and complement evasion. The present manuscript represents a tremendous piece of work and contains novel information about the complexity of the underlying processes that facilitate orchestration of cell division and encapsulation of newly formed daughter cells in streptococci. Although individual aspects have been studied before, the study provides impressively broad insight into the complex situation which involves different enzyme machineries and aims to dissect the entirety based on comprehensive experiments. However, there are some relevant aspects that should be addressed before publication to strengthen the authors conclusions.

Major points:

1) Line 159 ff: The authors state that "CpsC is likely recruited by a late-division protein downstream of FtsW and Pbp2x". This appears to be simple. Depletion of one of the late stage proteins does not only effect the composition of the late divisome complex, but lacking the enzymatic functions of FtsW (transglycosylase and putative flippase) and Pbp2x (transpeptidase) should in addition result in reduced PG synthesis. Whether CpsC is recruited via direct interaction with an individual downstream protein cannot be concluded from the experiments. This statement has to be toned down and the cellular complexity of the underlying mechanistic aspects and details need to be addressed.

2) Line 182 ff: Nakamoto et al work out the impact of each capsule biosynthesis protein on the localization of CpsA, CpsC, CpsJ and CpsH. However, this reviewer feels that results for complementation should be at least exemplarily provided for selected CPS biosynthesis genes (e.g. CpsJ localization in Δ cpsA + cpsA or Δ cpsI + cpsI or CpsH localization in Δ cpsD + cpsD) to provide evidence that indeed lack of protein-protein interactions are the cause of delocalization and neither downstream effects or regulatory alterations provoke the observations made. CpsC recruits CpsA and CpsH, localization of later CPS biosynthesis proteins is not only dependent on CpsC localization as shown in Fig. 3, e.g. CpsJ is more dependent on CpsA than CpsC itself. The concluding sentence should be reworded.

3) Line 200 ff: The authors state that septal PG synthesis is largely abolished, but there is no effect on CPS synthesis positioning and the amount of CPS produced. Based on the results shown in Suppl Fig. 6C is conceivable that there is an impact on capsule formation. The amount of CPS bound to cell wall and detected in the supernatant seems to be reduced. Since methicillin affects septal PG synthesis, the ratio of PG and CPS material present under the conditions chosen should be analyzed and quantified. It is explicitly stated in this paragraph that "PBP2x is required to recruit CpsC", which is contradictory to the conclusions in line 159 ff (major point 1).

4) Line 236 ff: Only one of the newly formed daughter cells seems to produce CPS although CpsC is localized at the poles of both (Fig. 5D). What is the author's explanation for this finding?

5) Line 310 ff: Nakamoto et al state that "Since the CPS complex at the cell pole remains functional, CPS can be installed without active PG synthesis, implying unlike WTA, nascent PG is not a required substrate." What is their explanation, how long chain CPS is transferred through multiple layers of peptidoglycan? Does a non-diffusive signal exclude that CPS is still bound to the lipid carrier, which is available due to the blockage of PG biosynthesis? As mentioned in major point 3, the amount of PG should be related to the amount of CPS produced/bound to PG as CPS seems to be overexpressed in the cells depicted in Suppl Fig. 9B.

Specific comments:

Line 37 ff: The authors refer to outer membrane synthesis in *E. coli*. Rausch et al (cited at a later point in the manuscript) studied the orchestration of PG synthesis and capsule formation in *Staphylococcus aureus* and this reference should be mentioned here providing information about a much closer situation than OM synthesis in a Gram-negative species.

Line 45 ff: The text reads as if CpsCD would organize PG synthesis. Ref 10 states "Wzd/Wze proteins may act as spatial regulators of capsule metabolism, ensuring the CPS is produced synchronously with the synthesis of peptidoglycan at the division septum". This should be rephrased.

Line 56: Bacterial tyrosine kinases and serine-threonine kinases are not related.

Line 65/163 ff: Are the cognate bPBPs which interact with FtsW and RodA known and was the respective partner protein of FtsW tested for its impact on CpsC localization?

Line 86 ff: The C-terminus of CpsC has been shown to interact with the kinase CpsD and to be important for signaling. Has the interaction of both proteins (in presence of the tag sfGFP/FLAG3) been studied in more detail? This reviewer feels that less CPS is produced (Supplementary Fig. 1B). Was a quantification of capsule material conducted? Why is the construct named CpsC-L-sfgfp? Is there a linker sequence? This is difficult to see from the information provided in the materials and method section.

Line 91 ff: The statement should be rephrased. The construct is expressed from the native cpsC locus and does not complement a cpsC deletion. See also comment on complementation above.

Line 157 ff and Fig S2/S4: How is the level of protein expressed and the rate of proteolysis determined? Additional bands that could represent degraded protein fractions are only detected for CpsA-FLAG3. Compared to the sfGFP construct and the other FLAG3 fusions, CpsC-FLAG3 seems to be overexpressed.

Line 219 ff: CpsC-L-sfGFP-H3H4 does not form polar foci as observed in the control. The authors should comment and explain.

Line 272 ff: PG and CPS are not really related. They use undecaprenyl phosphate and nucleotide-activated sugars as substrate and assemble the repeating units successively on the lipid carrier.

Line 279ff: It is unlikely that an integral membrane protein like CpsE “remains diffused in the cytosol”. It is also conceivable that the initial glycosyltransferase requires recruitment earlier than the flippase CpsJ.

Line 283 ff: CpsA-D are highly conserved in all serotypes whereas flippases as well as polymerases differ in substrate recognition, but also harbor conserved structures and features. Most likely interaction relies on these. Please comment.

Caption Fig. 3: Suppl Fig. 4 does not show representative micrographs. Please add or check this figure is correctly referenced.

Reviewer #1

The manuscript submitted by Nakamoto and colleagues is a refreshing and important manuscript that relates two processes crucial for bacteria – the synthesis of peptidoglycan and the synthesis of the surface capsular polysaccharide.

Essentially, authors propose that CpsC, an enzyme involved in the synthesis of the pneumococcal capsule, recruits the capsule synthetic machinery to the subcellular region where the capsule must be produced.

Authors not only confirmed previous observations related to the synthesis of CPS that had been made by other groups, but they integrated this knowledge into a model proposing that CpsC determines the localization of the capsule synthesis machinery.

Authors started by enquiring whether the localization of CpsC is dependent on proteins involved in the assembly of the division septum or on other proteins of the CPS operon.

Having settled that CpsC co-localizes with the divisome, and not with the elongosome, authors determined the order of the assembly of the capsule synthetic machinery, specifically, that CPS complex assembly starts with CpsCD, then CpsA and CpsH, the glycosyltransferases, and finally CpsJ.

Finally, authors claim the CpsCD is sufficient to direct the capsule synthesis as they were able to target CpsC to the cell pole and, consequently, reposition CPS synthesis in a way that is independent on the PGN synthesis.

Moreover, author confirmed that the “inability to synthesize CPS at the septum sensitizes the cell towards complement deposition, indicating that the location of the CPS complex could be important for virulence”.

I believe that this work is of great interest to those interested in understanding how bacteria assemble their cell surface in a way that keeps them protected from the infected host. Authors present evidence that supports their conclusions but should consider strengthening certain parts of their manuscript (see below).

We thank the reviewer for the detailed review and suggestions.

Major points

Point A.

Lines 91-92. Authors constructed a mutant that expresses CpsC-L-sfgfp encoded at the native locus and stated that “cpsC-L-sfgfp remained functional because it complemented cpsC (Supplemental Fig. 1A)”.

Could authors also include here the strain Delta cpsC // Delta cpsE // Pzn-cpsE so that the reader can observe, and compare, the effect exerted by the absence of a functional CpsC (or the presence of a non-functional one)?

I believe this is important as different groups have reported that the presence of a non-functional CpsC in encapsulated pneumococcal mutant strains may result in bacteria with capsule at the poles – a phenotype that authors have reported in the manuscript when they tried to direct capsule synthesis to the bacteria poles.

Thank you for your suggestions. We have performed spot dilution assays to include the suggested strains and CPS immunoblots (Fig. S1) with strains harboring \$\Delta\$ cpsC, a non-functional CpsC point mutation (*cpsC^{R121E}*), as well as tagged constructs of *cpsC* (i.e., *cpsC-L-sfgfp* and *cpsC-L-FLAG₃*). The GFP- and FLAG-tagged constructs are functional and support the production of full-length CPS.

Point B.

Lines 92-93. Author state that “...CPS production was restored to a normal level when Zn²⁺ was added to the medium (Supplementary Fig. 1B)”.

It is very difficult to confirm this statement as there are non-encapsulated bacteria in both strains shown in panels of Sup. Fig 1 after 60 min of induction with zinc.

Could authors use an additional way to demonstrate that the CPS production/full encapsulation is restored to normal levels when CpsC-L-sfGFP is produced instead of CpsC (authors could determine the percentage of fully encapsulated bacteria or compare the amount of CPS produced in equivalent number of bacteria).

It would also be important to demonstrate that the presence of zinc does not interfere with the production of capsule in the parental bacteria as the presence of different environmental cues may result in bacteria altered CPS production.

We have updated **Fig. S1** to include CPS IFM and immunoblots of WT, *cpsC-L-sfgfp*, *cpsC-L-FLAG3*. All of the strains tested are fully encapsulated with normal levels of CPS two hours after Zn^{2+} induction. The effect of Zn^{2+} on growth and CPS production has been reported in our previous work (Nakamoto *et al.*, 2021), and there is no detectable change in the CPS amount and growth defects when the concentration of Zn^{2+} is $\sim 0.5mM$.

Point C.

Line 96. "...confirmed that CPS synthesis, PG synthesis, and CpsC are co-localized (Fig. 1)."

It is difficult to analyze this figure 1 as it seems that the same color is being used to label CPS in the microscope image and in the demograph. This doesn't happen in the demographs associated with the localization of CpsC-L-sfgfp or the recently synthesized PG.

It would be helpful if authors used the same color code in all demographs in the manuscript and include the color/localization code in the figure.

Is it possible to associate a septal localization that seems to be observed in the demograph to a number? It could be the percentage of cells with mid cell localization. This would help to analyze data presented in demographs later in the manuscript (see below).

Additional information may be extracted from the demographs, namely by further analysing the patterns in short/younger? and longer/old? cells. It seems that PG incorporation precedes cpsC-L-sfGFP localization, which precedes CPS incorporation (which is in accordance with authors' claims).

We apologize for the confusion. The color scheme of the demographs is changed and they are now labeled with "septal" or "delocalised" (**Fig. 3, S3, and S5**). In addition, we agree with the reviewer that the demographs can provide information about the age of the cells. The text is modified "**As the length of the cell correlates with its age, PG incorporation apparently precedes CpsC recruitment**". Because there are many micrographs involved, we found it difficult to calculate the percentage of cells with mid-cell localized CpsC in an unbiased manner. We would like to argue that the demographs are sufficient to allow the readers to qualitatively evaluate whether the proteins are septally localized or not.

Point D.

Line 110-122. Authors have constructed mutants that lack different enzymes of the divisome and that express a fluorescently labelled CpsC. The observation of a signal scattered throughout the cell membrane was interpreted as delocalization.

Do authors know if the levels of delocalized CpsC-L-GFP are similar in the different mutants? If one of the mutations would lead to an increase in the amount of CpsC, then what it seems to be a delocalization could in fact be overexpression.

Authors could use the same procedure that they have used to show that CpsC-L-GFP is not cleaved (Figure S1C).

Thank you for the suggestion. As we validated the results of CpsC-L-GFP with CpsC-L-FLAG3, we performed Immunoblots to measure CpsC-L-FLAG3 levels instead. Their levels remained similar upon the depletion of *ftsZ*, *pbp2X*, and *ftsW* (**Fig. S3D**). Thus, delocalization of these proteins was not due to overexpression.

Point E.

Lines 157-159. "CpsC-L-FLAG3 was fully functional with minimal proteolysis (Supplemental Fig. 4), and it was localized to the septum like the sfGFP-tagged CpsC (Figs. 3, Supplemental Fig 3) "

Can authors show that the expression of Cps-L-FLAG3 results in bacteria that are fully encapsulated in the presence of zinc (to induce the expression of CpsE)?

As suggested, we showed that the expression of CpsC-L-FLAG3 resulted in bacteria that are fully encapsulated in the presence of Zn^{2+} (**Fig. S1**).

Point F.

Lines 201-203 "... (Supplemental Fig. 6A), CPS was still made in regions where FDAA was incorporated (Fig. 4A, Supplemental Fig. 6C). Consistently, methicillin treatment neither affects the positioning of CPS proteins (Fig. 4D) nor the amount and sizes of the CPS produced"

Could authors include some information relatively to the incorporation of capsule in the absence of methicillin? Do they have a recovery of fully encapsulated bacteria?

Agreed. The controls (no methicillin) are now shown in **Fig. S6C**. As expected, the bacteria are fully encapsulated.

Point G.

Lines 205-207 "These experiments indicate that although PBP2x is required to recruit CpsC, its transpeptidase activity does not affect the site selection of CPS synthesis."

How do authors know if *pbp2x* is inactivated?

The elongated cell phenotype shown in **Fig. 4A** and **Fig. 6A** is consistent with PBP2x inactivation because septal PG synthesis is inhibited (please refer to ref 58 and 59). The concentration of methicillin used in this study is based on these references, which demonstrated that PBP2x is specifically inhibited under this condition.

Point H.

Lines 220-222 "As expected, CpsC-L-sfGFP-H3H4 formed polar foci in the *popZ*⁺ background (Supplemental Fig. S8A), illustrating that the POLAR system robustly targets CpsC to the cell poles."

Authors should present data from *cpsC-H3H4* without *ipopZ*. If they are correct, and if *cpsC-h3h4* is functional, bacteria should be fully encapsulated.

As the reviewer has predicted, the strain harboring *cpsC-H3H4* is indeed fully encapsulated (**Fig. S8B**).

Point I.

Lines 237-240 "...with the capsule being produced at the poles, anti-CPS antibody staining resulted in foci that could be clearly distinguished from the nascent PG synthesized at the septum (Fig. 5D and supplemental Fig. 9B). Thus, we conclude that the mislocalized capsule complex remained functional and produced CPS outside of the divisome. "

To strengthen the claim that mislocalized CpsC is recruiting a functional capsule complex to places outside the divisome, authors should show that one of the other CPS proteins has also a polar localization in these strains expressing CpsC-L-H3H4 in the presence of *ipopZ*.

Thank you for the excellent suggestion. We have now included a new panel in **Fig. S8F** that shows that CpsJ-L-FLAG₃ was targeted to the cell pole in the *cpsC-H3H4 ipopZ* background.

Minor points

m1.

Line 138-139. Figure 2I.

Is the WT in this figure the Δ *kphA* mutant? To prevent misinterpretations, it may be useful to state clearly in the figure that one of the mutants is Δ *kphA* Δ *mreC* while the other is, I assume, Δ *kphA*

Changed.

m2.

Lines 180-181. "As expected, CpsA-L-FLAG₃, CpsJ-L-FLAG₃, and CpsH-L-FLAG₃ were intact and localized at the septum (Supplemental Fig. 4A and 5B)."

Do authors know if these constructs are functional? This information should be clearly stated.

We showed the FLAG-tagged CpsJ and CpsH are functional (**Fig. S1A and C**). It is difficult to assay the functionality of CpsA as it does not affect growth and CPS production, as reported by the Yother laboratory (PMC225014).

m3.

Lines 184-187. "We propose that CpsC-L-FLAG3 arrives at the divisome first, because Δ cpsC delocalized CpsA, CpsJ, and CpsH (Fig. 3, Supplemental Fig. 5B). Furthermore, CpsC localization is independent of other CPS factors, except CpsD, which was shown to stabilize CpsC (Fig. 3; Supplemental Fig. 5A)."

It is difficult to associate a demograph to a conclusion of "localized vs non-localized". It gets even more difficult as in Figure 3 there is no legend stating what is the meaning of different colors. Could authors include in each demograph whether the FLAG-labeled protein has a septal localization? Authors should take a particular look at Figure S5 panel A. One can observe in the microscope image that the deletion of CpsD interferes with septal localization of CpsC-L-FLAG3. However, the demograph seems like others where CpsC-L-FLAG3 has a clear septal localization. It would help the readers, if authors would associate a code to each demograph of Fig 3 where they would identify if the labeled protein has the expected septal localization.

We apologize for the confusion. Figures are updated to state whether proteins are "septal localised" or "delocalised" with a more consistent color scheme. We also added scale bars in **Fig. 3** to indicate the meaning of the color code, which represents signal intensity.

In Figure 3 and S5, were bacteria expressing capsule? I am assuming that these observations were done in the absence of zinc but it is not clear. Authors should state clearly that these localizations were carried out in the absence of capsule (as bacteria were propagated in BHI without zinc)

The figure legend of **Figs. 3 and S5** are modified to indicate the experiments were done in the absence of Zn^{2+} .

m4.

Lines 220-221. "As expected, CpsC-L-sfGFP-H3H4 formed polar foci in the popZ+ background (Supplemental Fig. S8A), illustrating that the POLAR system robustly targets CpsC to the cell poles. Could authors strengthen this claim by presenting the percentage of cells with CpsC at the poles in the different conditions?"

We have updated the figure legends to include the percentage of cells with CpsC-L-sfgfp at the poles: "NUS1783 [P_{spxB} -cpsC-L-sfgfp-H3H4] (10.19%) and NUS1862 [P_{spxB} -itag-popZ // P_{spxB} -cpsC-L-sfgfp-H3H4] (77.18%)."

m5.

Line 687 "were then immunostained with anti-FLAG and anti-serotype 2 capsule antibodies as described in..."

there is no capsule staining in this figure. Could authors correct the figure legend or include the reported data in the figure.

Corrected.

m6.

Line 690. "Shown are the demographs of the fluorescent channels depicting the location of the FLAG tag fusion proteins as well as the tentative recruitment pathway. Representative micrographs are presented in Supplemental Fig. 4. "

Could authors confirm this sentence? I believe they were referring to Sup. Fig. 5.

Corrected.

m7.

Lines 227-228. "However, when CpsC was targeted to the pole (i.e., strain NUS2029 [cpsC-H3H4 // PspxB-itag- popZ]), no CPS was detected at the septum (Fig. 5C). Instead, CPS was only produced at the poles. "

Is CpsC-H3H4 functional?

Yes, CpsC-H3H4 is functional and supports capsule production (**Fig. S8B**).

m8.

Lines 249-251 “As mentioned above, the bacteria that synthesize CPS at the poles as a result of the CpsC-H3H4 fusion did not have capsule at the septal region (Fig. 6A). Instead, C3b could bind to the septal region as can be seen in Fig. 6A, “

Could authors use other color to label CPS? In a dark background it is very difficult to see red signals. It seems that the bacteria cpsC-H3H4 // PspxB-ipopZ are fully encapsulated.

Corrected.

m9

Line 667 and in other locations of the manuscript.

Fluorescence microscopy instead of fluorescent microscopy?

Corrected.

Reviewer #2

The work submitted by Nakamoto et al “The divisome but not the elongasome organizes capsule synthesis in *Streptococcus pneumoniae*” studies the spatiotemporal orchestration of capsule synthesis with peptidoglycan biosynthesis and cell division. The authors showed that CPS synthesis initiates from the division septum and that assembly of the synthesis complex is organized by the bacterial tyrosine kinase system CpsCD. Recruitment of the proteins to the division septum is associated with the divisome, but not the elongasome. Using the POLAR system, CPS synthesis can be repositioned to the cell pole resulting in diplococci that lack CPS at the septum. Nakamoto et al propose the importance of septal CPS for chain formation and complement evasion.

The present manuscript represents a tremendous piece of work and contains novel information about the complexity of the underlying processes that facilitate orchestration of cell division and encapsulation of newly formed daughter cells in streptococci. Although individual aspects have been studied before, the study provides impressively broad insight into the complex situation which involves different enzyme machineries and aims to dissect the entirety based on comprehensive experiments. However, there are some relevant aspects that should be addressed before publication to strengthen the authors conclusions.

We thank the reviewer for the detailed review and suggestions.

Major points:

1) Line 159 ff: The authors state that “CpsC is likely recruited by a late-division protein downstream of FtsW and Pbp2x”. This appears to simple. Depletion of one of the late stage proteins does not only effect the composition of the late divisome complex, but lacking the enzymatic functions of FtsW (transglycosylase and putative flippase) and Pbp2x (transpeptidase) should in addition result in reduced PG synthesis. Whether CpsC is recruited via direct interaction with an individual downstream protein cannot be concluded from the experiments. This statement has to be toned down and the cellular complexity of the underlying mechanistic aspects and details need to be addressed.

Agreed. The statement is toned down to: “Thus, CpsC recruitment requires late divisome proteins in the divisome”.

2) Line 182 ff: Nakamoto et al work out the impact of each capsule biosynthesis protein on the localization of CpsA, CpsC, CpsJ and CpsH. However, this reviewer feels that results for complementation should be at least exemplarily provided for selected CPS biosynthesis genes (e.g. CpsJ localization in Δ cpsA + cpsA or Δ cpsI + cpsI or CpsH localization in Δ cpsD + cpsD) to provide evidence that indeed lack of protein-protein interactions are the cause of delocalization and neither downstream effects or regulatory alterations provoke the observations made. CpsC recruits CpsA and CpsH, localization of later CPS biosynthesis proteins is not only dependent on CpsC localization as shown in Fig. 3, e.g. CpsJ is more dependent on CpsA than CpsC itself. The concluding sentence should be reworded.

Thank you for the excellent suggestion. We have complemented *cpsC*, *cpsT*, *cpsG* and *cpsI* and showed that the septal localization of CpsA, CpsJ, and CpsH was restored (Fig. S5C).

3) Line 200 ff: The authors state that septal PG synthesis is largely abolished, but there is no effect on CPS synthesis positioning and the amount of CPS produced. Based on the results shown in Suppl Fig. 6C is conceivable that there is an impact on capsule formation. The amount of CPS bound to cell wall and detected in the supernatant seems to be reduced. Since methicillin affects septal PG synthesis, the ratio of PG and CPS material present under the conditions chosen should be analyzed and quantified. It is explicitly stated in this paragraph that “PBP2x is required to recruit CpsC”, which is contradictory to the conclusions in line 159 ff (major point 1).

While we share the same view with the reviewer, we could not detect any statistical significant differences in the amount of CPS in cells with or without methicillin treatment (**Fig. S6B**). We also measured the area of the new PG synthesized (**Fig. S6D**) and compared it with the new CPS produced, but we could not detect any significant difference. Thus, the PBP2x protein, but not its transpeptidase activity, is required for the recruitment of CpsC. We hope this point is communicated after we addressed the reviewer’s concern regarding line 159.

4) Line 236 ff: Only one of the newly formed daughter cells seems to produce CPS although CpsC is localized at the poles of both (Fig. 5D). What is the author’s explanation for this finding?

The cells shown in **Fig. 5D** are likely just divided. Thus, the CPS can only be found in a single pole. Also, the experiment was done after a very short pulse of the inducer. We think CpsC-H3H4 can be in one or both poles, similar to the controls shown in **Fig. S8** and the observation by the Shapiro laboratory (PMID: 20149103).

5) Line 310 ff: Nakamoto et al state that “Since the CPS complex at the cell pole remains functional, CPS can be installed without active PG synthesis, implying unlike WTA, nascent PG is not a required substrate.” What is their explanation, how long chain CPS is transferred through multiple layers of peptidoglycan? Does a non-diffusive signal exclude that CPS is still bound to the lipid carrier, which is available due to the blockage of PG biosynthesis? As mentioned in major point 3, the amount of PG should be related to the amount of CPS produced/bound to PG as CPS seems to be overexpressed in the cells depicted in Suppl Fig. 9B.

In light of the reviewer’s comments, we have modified our statements to reduce certainty:

“Since the CPS complex remains functional at the cell pole, it implies that unlike WTA, CPS can be installed on mature PG. Consistently, disruption of septal PG synthesis by methicillin treatment did not affect CPS synthesis (Supplemental Fig. S6). How the long CPS chains can be transferred to the mature PG warrants further investigation. While we did not exclude the possibility that there are other cell surface anchors for CPS, the non-diffusive signal of the capsule and the lack of growth phenotype suggest CPS has been transferred to the PG and is no longer attached to the Und-P (Fig. 5 and supplemental Fig. S9).”

A non-diffusive signal is consistent with the CPS being anchored to the PG because we did not see fast lateral diffusion that suggests a lipid anchor. Nevertheless, while unlikely, we could not exclude the possibility that the CPS can be attached elsewhere on the envelope. The mechanism of transferring the capsule chains to the mature PG is still unknown and is beyond the scope of this study.

We also noticed the brighter CPS signal in the two cells shown in **Fig. S9B**. Yet, there was no detectable change in the CPS amount in this strain as judged by immunoblotting and ELISA (**Fig. S8C and S8D**). Thus, we refrain from concluding CPS is overexpressed when the CPS complex is redirected to the poles.

Specific comments:

Line 37 ff: The authors refer to outer membrane synthesis in *E. coli*. Rausch et al (cited at a later point in the manuscript) studied the orchestration of PG synthesis and capsule formation in *Staphylococcus aureus* and this reference should be mentioned here providing information about a much closer situation than OM synthesis in a Gram-negative species.

We cited the *E. coli* study here because it investigated the cell biology aspect of the coordination, which matches well with this study. The biochemical study from Rausch et al found that PknB negatively regulates CPS synthesis likely by phosphorylating CPS proteins. As we did not investigate StkP, it is less related. A line is added to acknowledge the contribution of Rausch et al. (line 42):

“A link between CPS and PG synthesis was suggested because the serine-threonine kinase PknB (or StkP) negatively regulates CPS synthesis in *Staphylococcus aureus*”.

Line 45 ff: The text reads as if CpsCD would organizes PG synthesis. Ref 10 states “Wzd/Wze proteins may act as spatial regulators of capsule metabolism, ensuring the CPS is produced synchronously with the synthesis of peptidoglycan at the division septum”. This should be rephrased.

Corrected.

Line 56: Bacterial tyrosine kinases and serine-threonine kinases are not related.

Corrected.

Line 65/163 ff: Are the cognate bPBPs which interact with FtsW and RodA known and was the respective partner protein of FtsW tested for its impact on CpsC localization?

Yes, the interacting partner of FtsW and RodA are Pbp2x and Pbp2b respectively (Taguchi, *et al.* 2019). FtsW, RodA, Pbp2x, and Pbp2b were tested for their impact on CpsC localization in this study (**Fig. 2**).

Line 86 ff: The C-terminus of CpsC has been shown to interact with the kinase CpsD and to be important for signalling. Has the interaction of both proteins (in presence of the tag sfGFP/FLAG3) been studied in more detail? This reviewer feels that less CPS is produced (Supplementary Fig. 1B). Was a quantification of capsule material conducted? Why is the construct named CpsC-L-sfgfp? Is there a linker sequence? This is difficult to see from the information provided in the materials and method section.

Thank you for pointing this out. We have included a CPS immunoblot in **Fig. S1** to show the amount of CPS is similar between the wild-type and strains expressing *cpsC-L-sfgfp-L-FLAG₃*. The mutants are fully encapsulated. “L” refers to a linker sequence (GSAGSAAGSG) and we have included the information in our updated materials and methods.

Line 91 ff: The statement should be rephrased. The construct is expressed from the native *cpsC* locus and does not complement a *cpsC* deletion. See also comment on complementation above.

Rephrased.

Line 157 ff and Fig S2/S4: How is the level of protein expressed and the rate of proteolysis determined? Additional bands that could represent degraded protein fractions are only detected for CpsA-FLAG3. Compared to the sfGFP construct and the other FLAG3 fusions, CpsC-FLAG3 seems to be overexpressed.

The statement is reworded to “no significant FLAG-tag cleavage detected” to avoid confusion. The amount of *cpsC-L-FLAG₃* and *cpsC-L-sfgfp* cannot be compared because they are measured in different blots and are detected by different antibodies. We included immunoblots in **Fig. S4** to illustrate the FLAG tags were not significantly cleaved. Antibodies against CpsC are not available because we could not purify CpsC. All we could do at the moment is to place the cassette at the native locus so that its expression will be driven by the native promoter and RBS.

Line 219 ff: CpsC-L-sfGFP-H3H4 does not form polar foci as observed in the control. The authors should comment and explain.

The reason why the CpsC-L-sfGFP-H3H4 signal is more dispersed is unclear. We speculate CpsC may form oligomers, leading to more diffusive signals at the poles. This is beyond the scope of this study and therefore we are not further pursuing it.

Line 272 ff: PG and CPS are not really related. They use undecaprenyl phosphate and nucleotide-activated sugars as substrate and assemble the repeating units successively on the lipid carrier.

Corrected as suggested.

Line 279ff: It is unlikely that an integral membrane protein like CpsE “remains diffused in the cytosol”. It is also conceivable that the initial glycosyltransferase requires recruitment earlier than the flippase CpsJ.

Agreed and the text is changed. There is no evidence that suggests the localization of CpsJ is dependent on CpsE, as it is septal localized in the wild-type and $\Delta cpsK/\Delta cpsL/\Delta cpsP$ backgrounds. We cannot comment on the localization of CpsE (the initiating GT) with certainty, as we could not generate FLAG or GFP fusions of it.

Line 283 ff: CpsA-D are highly conserved in all serotypes whereas flippases as well as polymerases differ in substrate recognition, but also harbor conserved structures and features. Most likely interaction relies on these. Please comment.

Possibly. At this moment, we do not have any strong experimental evidence to prove or disprove this hypothesis. Sequence alignments and AlphaFold predictions have yet to identify the conserved structures and features that generate testable hypotheses, as many of these regions are seemingly important for their enzymatic function. Thus, we will keep this line of investigation for future studies.

Caption Fig. 3: Suppl Fig. 4 does not show representative micrographs. Please add or check this figure is correctly referenced.

Corrected.

Reviewer #1 (Remarks to the Author):

Authors have included most of the requested information, addressed this reviewer's concerns and have strengthened the interpretation of the observed results.

I believe this manuscript has been improved and that it will be of interest to the scientific community interested in understanding how bacteria assemble their cell surface in a way that keeps them protected from the infected host.

Reviewer #2 (Remarks to the Author):

The manuscript "The divisome but not the elongasome organizes capsule synthesis in *Streptococcus pneumoniae*" submitted by Nakamoto et al studies the spatiotemporal orchestration of capsule synthesis with peptidoglycan biosynthesis and cell division. In the revised manuscript the authors addressed relevant aspects, performed additional experiments (i.e. complementation of *cpsC*, *cpsT*, *cpsG* and *cpsI* that restored septal localization of CpsA, CpsJ, and CpsH) which further strengthen the authors conclusion and the manuscript in general. My concerns have been addressed and I support publication.